# Synthetic Data (Almost) from Scratch: Generalized Instruction Tuning for Language Models

## Abstract

We introduce *Generalized Instruction Tuning* (called GLAN), a general and scalable method for instruction tuning of Large Language Models (LLMs). Unlike prior work that relies on seed examples or existing datasets to construct instruction-tuning data, GLAN exclusively utilizes a pre-curated taxonomy of human knowledge and capabilities as input and generates large-scale synthetic instruction data across all disciplines. Specifically, inspired by the systematic structure in human education system, we build the taxonomy by decomposing human knowledge and capabilities to various fields, sub-fields and ultimately, distinct disciplines semi-automatically, facilitated by LLMs. Subsequently, we generate a comprehensive list of subjects for every discipline and proceed to design a syllabus tailored to each subject, again utilizing LLMs. With the fine-grained key concepts detailed in every class session of the syllabus, we are able to generate diverse instructions with a broad coverage across the entire spectrum of human knowledge and skills. Extensive experiments on large language models (e.g., Mistral) demonstrate that GLAN excels in multiple dimensions from mathematical reasoning, coding, academic exams, logical reasoning to general instruction following without task-specific training data. In addition, GLAN allows for easy customization and new fields or skills can be added by simply incorporating a new node into our taxonomy.

## 1 Introduction

Large Language Models (LLMs) have enabled unprecedented capabilities to understand and generate text like humans. By scaling up model size and data size (Kaplan et al., 2020; Hoffmann et al., 2022), LLMs are better at predicting next tokens and prompting to perform certain tasks with a few demonstrations (Brown et al., 2020). However, these capabilities do not directly translate to better human instruction-following ability (Ouyang et al., 2022). Instruction tuning (Wei et al., 2022) bridges this gap by fine-tuning LLMs on instructions paired with human-preferred responses.

Previous work has constructed instruction tuning data using seed examples or existing datasets (Xu et al., 2023a; Wang et al., 2023). For example, FLAN (Wei et al., 2022) aggregates traditional NLP datasets into an instruction-following set. However, the availability of only a few thousand NLP tasks (Wang et al., 2022; Longpre et al., 2023) restricts the generalization capabilities of LLMs trained on FLAN (Xu et al., 2023a). Recently, the Self-instruct approach (Wang et al., 2023) has generated synthetic instruction tuning datasets from a limited pool of human-written seed instructions. Evolve-Instruct (Xu et al., 2023a) further enhances this by augmenting existing instruction tuning datasets through rewriting operations using LLMs. Despite these advancements, these strategies primarily rely on data augmentation, meaning the range of domains or tasks covered by the augmented datasets remains constrained by the original input datasets.

How to create a *general* instruction tuning dataset? We draw inspiration from the systematic structure in human education system. The structure of human education includes several levels, starting from early childhood education up to higher education and beyond (Wikipedia contributors, 2023). Within each level, a student acquires knowledge, skills, and values in a systematic process. The courses a student learns from primary school to college cover a broad range of knowledge and skills,

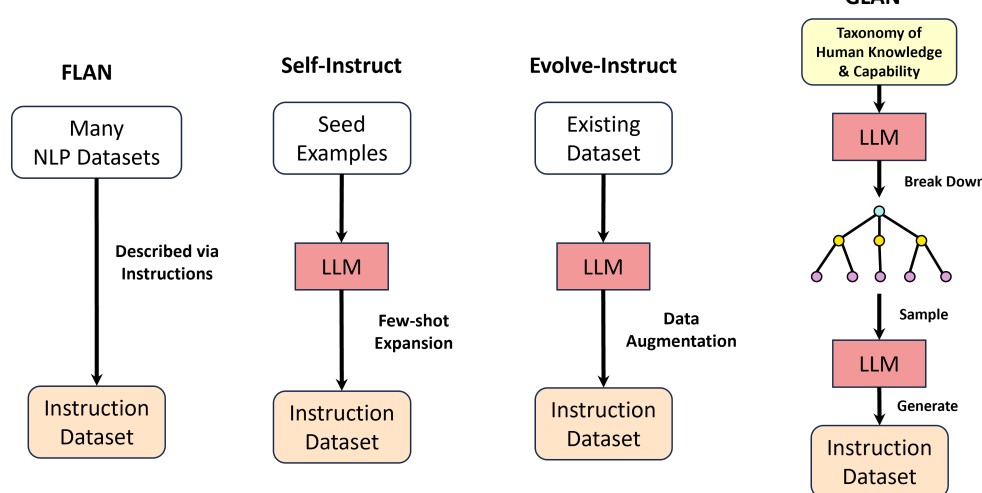

Figure 1: Comparing GLAN with FLAN, Self-Instruct and Evolve-Instruct. The inputs of FLAN, Self-Instrct and Eovlve-Instruct are either seed examples or existing datasets, which limits the scope of domains of instructions that these methods can generate. GLAN takes the taxonomy of human knowledge & capabilities as input to ensure the broad coverage of generated instructions in various domains. This taxonomy is then broken down into smaller pieces and recombined to generate diverse instruction data.

which facilitates the development of a diverse array of abilities. We believe that the systemic framework of the human education system has the potential to help the generation of high-quality and *general* instruction data, which spans a diverse range of disciplinary areas.

In this paper, we introduce a generalized instruction tuning paradigm GLAN (shorthand for **G**eneralized Instruction-Tuning for **L**arge **LAN**guage Models) to generate synthetic instruction tuning data almost from scratch. As shown in Figure 1, unlike existing work (Xu et al., 2023a; Luo et al., 2023b;a; Mukherjee et al., 2023), GLAN exclusively utilizes a pre-curated taxonomy of human knowledge and capabilities as input and generates large-scale instruction data systematically and automatically across all disciplines. Specifically, inspired by the structure of the human education system, the input taxonomy is constructed by decomposing human knowledge and capabilities to various fields, sub-fields, and, ultimately, distinct disciplines semi-automatically, facilitated by LLMs and human verification. The cost of human verification process is low due to the limited number of disciplines in the taxonomy. As shown in Figure 1, we then further break down these disciplines into even smaller units. We continue to generate a comprehensive list of subjects for every discipline and proceed to design a syllabus tailored to each subject, again utilizing LLMs. With the fine-grained key concepts detailed in every class session of the syllabus, we can first sample from them and then generate diverse instructions with broad coverage across the entire spectrum of human knowledge and skills. The process described above mirrors the human educational system, where educators in each discipline craft a series of subjects for student learning. Instructors then develop a syllabus for each subject, breaking down the content into specific class sessions. These sessions are then further divided into core concepts that students must comprehend and internalize. Based on these detailed core concepts outlined in the syllabus, teaching materials and exercises are subsequently created, which are our instruction tuning data.

GLAN is general, scalable and customizable. GLAN is a general method, which is task-agnostic and is capable of covering a wide range of domains. GLAN is scalable. Similar to Wang et al. (2023); Xu et al. (2023a), GLAN generates instructions using LLMs, which can produce instructions on a massive scale. Moreover, the input of GLAN is a taxonomy, which is generated by prompting an LLM and human verification, requiring minimal human effort. GLAN allows for easy customization. New fields or skills can be added by simply incorporating a new node into our taxonomy. Note that each node of the taxonomy can be expanded independently, which means that we only need to apply our method to the newly added nodes without re-generating the entire dataset. Extensive experiments on large language models (e.g., Mistral) demonstrate that GLAN excels in

multiple dimensions from mathematical reasoning, coding, academic exams, and logical reasoning to general instruction following without using task-specific training data of these tasks.

## 2 GLAN: GENERALIZED INSTRUCTION-TUNED LANGUAGE MODELS

GLAN aims to create synthetic instruction data covering various domains of human knowledge and capabilities on a large scale. As shown in Algorithm 1, we first build a taxonomy of human knowledge and capabilities using frontier LLMs (i.e., GPT-4) and human verification. The taxonomy naturally breaks down human knowledge and capabilities to *fields*, *sub-fields*, and ultimately different *disciplines* (see Section 2.1). The following steps are fully autonomously facilitated by GPT-4 (or GPT-3.5). Then for each discipline, we again instruct GPT-4 to further decompose it into a list of subjects within this discipline (Section 2.2). Similar to an instructor, GPT-4 continues to design a syllabus for each subject, which inherently breaks a subject into various class sessions with key concepts that students need to master (Section 2.3). With the obtained class sessions and key concepts, we are ready to construct synthetic instructions. We prompt GPT-4 to generate homework questions based on randomly sampled class sessions and key concepts as well as the syllabus (Section 2.4). We recursively decompose human knowledge and capabilities into smaller units until atomic-level components (i.e., class sessions and key concepts). We expect to randomly combine these class sessions and key concepts to ensure the coverage and diversity of synthetic instructions.

---
**Algorithm 1** GLAN Instruction Generation

---
$\mathbb{D} \leftarrow$ build_taxonomy()   ▷ build a taxonomy and return a list of *disciplines* (Section 2.1)
$\mathbb{L} \leftarrow \varnothing$
**for** each discipline $d \in \mathbb{D}$ **do**
    $\mathbb{S} \leftarrow$ generate_subjects($d$)       ▷ Obtain a list of *subjects* in $d$ (Section 2.2)
    **for** each subject $s \in \mathbb{S}$ **do**
        $\mathcal{A} \leftarrow$ generate_syllabus($s, d$)       ▷ Return syllabus $\mathcal{A}$ for $s$ (Section 2.3)
        $\mathbb{C}, \mathbb{K} \leftarrow$ extract_class_details($\mathcal{A}$)   ▷ Extract class sessions and key concepts
(Section 2.3)
        $\mathbb{Q} \leftarrow$ generate_instructions($\mathcal{A}, \mathbb{C}, \mathbb{K}, d$)   ▷ Generate instructions by sampling
class sessions and key concepts (Section 2.4)
        $\mathbb{L} \leftarrow \mathbb{L} \cup \mathbb{Q}$
    **end for**
**end for**
**return** $\mathbb{L}$

---

### 2.1 TAXONOMY OF HUMAN KNOWLEDGE AND CAPABILITIES

We build a taxonomy of human knowledge and capabilities to guide the generation of synthetic instructions. Therefore, its coverage is important. On the other hand, it is also essential to make the taxonomy highly extensible, since the preferred capabilities of LLMs may change over time. In the first step, we propose to generate the taxonomy by prompting GPT-4 with a set of different instructions (e.g., list all fields of human knowledge and capabilities). Then, we do human post-editing to ensure its correctness and completeness. Due to the limited number of fields, sub-fields, and disciplines in our taxonomy, the cost of human verification is reasonably low. Another advantage of human post-editing is that we can easily add new fields or disciplines to the taxonomy as needed.

Our taxonomy currently covers a diverse range of knowledge and capabilities in both academic education and vocational training. The top level of the taxonomy contains *fields* such as *Natural Sciences*, *Humanities*, or *Services* (vocational training). These fields branch out to various *sub-fields* and/or *disciplines* such as *Chemistry*, *Sociology* or *Retailing*. We keep breaking down nodes of the taxonomy until *disciplines*, and we leave the breaking down of disciplines to automatic methods described in the following sections. By collecting the leaf nodes of the taxonomy, we obtain a list of disciplines $\mathbb{D} = \{d_1, d_2, \ldots, d_M\}$.

## 2.2 Subject Generator

As in Algorithm 1, for each discipline $d$, we aim to extract the list of subjects in it through prompt engineering. Specifically, we instruct `GPT-4` to `act as an education expert of discipline` $d$ `and design a list of subjects a student should learn.` The completion of `GPT-4` contains a comprehensive list of subjects and their meta data (e.g., level, introduction and subtopics of the subject) in unstructured text format, which can not be directly used in subsequent steps. We therefore used another round of prompting to convert the completion to `JSONL` format:

> **Prompt**
>
> Transform the above to JSONL format so that it is easier for a computer to understand. Enclose the JSONL output between two sets of triple backticks. For each JSONL object, use the keys "subject_name", "level" and "subtopics".

It is worth noting that generating a subject list in `JSONL` format using a single prompt is feasible. However, we refrain to do so, because we observe that incorporating additional formatting instructions directly into the prompt can compromise the quality of the resulting subject list. These extracted subjects (as well as their meta data) $\mathbb{S} = \{s_1, s_2, \ldots, s_N\}$ can be subsequently used in next steps. For each $s \in \mathbb{S}$, let `s.name`, `s.level` and `s.subtopics` denote the name, grade level and subtopics of subject $s$, respectively. We can apply the above prompts multiple times to ensure better coverage of subjects within this discipline.

## 2.3 Syllabus Generator

For each subject $s$, we have already extracted its name (`s.name`), grade level (`s.level`), and a small set of included sub-topics (`s.subtopics`) in a structured format. In this section, we aim to further segment each subject into smaller units, making them more suitable for creating homework assignments. We consult `GPT-4` to design a syllabus for this subject. We opt for syllabus generation for the following reasons. Firstly, a syllabus essentially breaks down the main topic of a subject into smaller segments in a hierarchical manner. Specifically, each subject comprises several class sessions, and each session covers a variety of sub-topics and key concepts. Secondly, a syllabus provides an introduction, objectives, and expected outcomes of a subject, which are inherently useful for formulating homework questions. We instruct `GPT-4` to 1) design a syllabus based on its meta data (`s.level`, `s.name` and `s.subtopics`); 2) break the subject into different class sessions; 3) provide details for each class session with a description and detailed key concepts students need to master.

Let $\mathcal{A}$ denote the generated syllabus. The resulting syllabus $\mathcal{A}$ is in unstructured text format. However, class session names and key concepts of each class are required in the instruction generation step (see Algorithm 1). Similar to the process of subject list extraction in Section 2.2, we again extract these meta data of each class session by prompting `GPT-4`. As a result, we obtain a list of class sessions $\mathbb{C} = \{c_1, c_2, \ldots, c_{|\mathbb{C}|}\}$ and their corresponding key concepts $\mathbb{K} = \{\mathbf{k}_1, \mathbf{k}_2, \ldots, \mathbf{k}_{|\mathbb{C}|}\}$. The detailed prompt for syllabus generation is in Appendix A.3.

## 2.4 Instruction Generator

Given a syllabus $\mathcal{A}$ as well as a list of its class sessions $\mathbb{C}$ and their associated key concepts $\mathbb{K}$, we are ready to generate homework questions and their answers. To generate diverse homework questions, we first sample one or two class session names from $\mathbb{C}$ and one to five key concepts under these selected class sessions. Let $\hat{\mathbb{C}}$ denote the selected class session names and $\hat{\mathbb{K}}$ the selected key concepts. Then we prompt `GPT-4` (or `GPT-3.5`) to generate a homework question given the selected class sessions $\hat{\mathbb{C}}$ and key concepts $\hat{\mathbb{K}}$ as well as the syllabus $\mathcal{A}$. We intend to give `GPT-4/3.5` more context (e.g., what students have already learned in previous sessions) when creating assignments. Therefore, we additionally instruct GPT to consider that students have learned up to class sessions $\hat{\mathbb{C}}$ when crafting homework and try to leverage multiple key concepts across different class sessions. See details of our prompt for instruction generation in Appendix A.4.

**Sampling Class Sessions and Key Concepts** In a single syllabus, there are numerous class sessions and key concepts. We have two strategies to sample from them. In the first strategy, we gener-

ate assignments from a single class session. Therefore, we have only one class session name. Suppose we have $m$ key concepts in total in this session. We randomly sample one to five key concepts from the $m$ key concepts, which means we have totally $\sum_{i=1}^{5} \binom{m}{i}$ unique combinations. In this strategy, we focus on creating *basic* homework questions. To make the resulting questions more challenging (combine knowledge from multiple class sessions), we propose a second strategy to combine key concepts from two class sessions in the second strategy. We intend to generate questions leverage knowledge from two different class sessions. Suppose we have $m_1$ and $m_2$ key concepts in the first and second class sessions, respectively. We can have $\sum_{i=2}^{5} \binom{m_1+m_2}{i} - \sum_{i=2}^{5} \binom{m_1}{i} - \sum_{i=2}^{5} \binom{m_2}{i}$ different combinations, which is significantly more than that of the first strategy. We use both strategies to ensure our created questions are diverse in difficulty levels.

**Answer Generation**    After we generate questions in previous steps, we simply send these questions to `GPT-3.5` and collect answers. We use `GPT-3.5` for answer generation, because we find the quality of generated answers from `GPT-3.5` is sufficiently good and using `GPT-3.5` is significantly faster than `GPT-4`. The resulting question-answer pairs are our instruction tuning data. With a huge amount of question-answer pairs ranging from different disciplines with various difficulty levels, we expect the resulting LLM can excel in a wide range of tasks.

## 3 EXPERIMENTS

### 3.1 DATA GENERATION

**Taxonomy Creation**    By asking `GPT-4` to create a taxonomy of human knowledge and capabilities, we end up with a set of fields, sub-fields, and disciplines that cover a broad range of domains in human knowledge and capabilities. Next, we ask human annotators to decide whether these elements in the taxonomy should be kept or not in order to reduce the redundancy of the taxonomy while maintaining its correctness. Note that if a field or sub-field is marked as *remove*, we remove its descendant as well. We kept 126 *disciplines* after majority voting (provided in supplementary materials). Note that it is feasible to manually add extra disciplines, sub-fields, or fields whenever necessary.

**Subject and Syllabus Generation**    During the subject list and syllabus generation, we prompt `GPT-4` and employ nucleus sampling (Holtzman et al., 2020) with temperature $T = 1.0$ and top-$p = 0.95$ to encourage diversity. We do not use `GPT-3.5-turbo` since some subjects belong to the long-tail distribution which may not be effectively modeled by `GPT-3.5-turbo`. To ensure diversity and completeness of the generated subjects, we query `GPT-4` 10 times for each discipline (Section 2.2). There are 100 to 200 subjects for each discipline on average. It is worth noting that the same subjects may appear in different disciplines. For instance, the subject *calculus* is both in physics and mathematics. We do not de-duplicate those subjects, since it may reflect their importance in human knowledge. Given a subject in a specified discipline, we query `GPT-4` for only one time to design a syllabus (see details in section 2.3). The temperature and top-$p$ are still set to 1.0 and 0.95, respectively. The number of class sessions contained in each syllabus varies from 10 to 30 and each class session contains around five key concepts.

**Instruction Data Generation**    Each instruction data consists of a question and its answer. We choose to generate questions and answers separately since we observed that separate generations lead to higher quality outputs. After question generation with `GPT-4`, each question is then answered by `GPT-3.5-turbo` with temperature $T = 0.7$, top-$p = 0.95$ (we use a lower temperature in order to make the resulting answers more accurate). We use `GPT-3.5-turbo` instead of `GPT-4` for answer generation, because `GPT-3.5-turbo` is significantly faster with reasonably good results. According to the calculation method outlined in Section 2.4, we have over 500 million unique combinations of class sessions and key concepts, which guarantees the diversity of the generated data. In this paper, we generate 10 million instruction-response pairs in total and then we do training data decontamination. Specifically, the training instruction-response pairs are decontaminated by removing pairs that contain questions or input prompts from the test and training (if any) sets of benchmarks we evaluate. We exclude the training set of benchmarks we evaluate to verify the generalization capability of our synthetic data.

Table 1: Main results on Mathematical Reasoning, Coding, Logical Reasoning, and Academic Exam benchmarks. Best results are in boldface, while the second best results are underscored.

| Model | $\theta$ | HumanE | MBPP | GSM8K | MATH | BBH | ARC-E | ARC-C | MMLU |
|---|---|---|---|---|---|---|---|---|---|
| GPT-4 | – | 88.4 | 80.0 | 92.0 | 52.9 | 86.7 | 95.4 | 93.6 | 86.4 |
| GPT-3.5-turbo | – | 72.6 | 70.8 | 74.1 | 37.8 | 70.1 | 88.9 | 83.7 | 70.0 |
| LLaMA2 | 7B | 12.8 | 36.2 | 15.4 | 4.2 | 39.6 | 74.6 | 46.3 | 45.9 |
| Orca 2 | 7B | 17.1 | 28.4 | 55.7 | 10.1 | 42.8 | 87.8 | 78.4 | 53.9 |
| WizardLM v1.2 | 13B | 31.7 | 47.9 | 46.8 | 9.0 | 48.4 | 74.2 | 50.2 | 52.7 |
| Mistral | 7B | 28.0 | 50.2 | 43.4 | 10.0 | 56.1 | 79.5 | 53.9 | 62.3 |
| Mistral Instruct | 7B | 46.7 | 31.7 | 24.4 | 8.2 | 46.0 | 76.9 | 52.0 | 53.7 |
| MetaMath Mistral | 7B | 35.4 | 48.6 | 77.7 | 28.2 | 55.7 | 77.3 | 51.0 | 61.0 |
| WizardMath v1.1 | 7B | **51.2** | 54.1 | **83.2** | **33.0** | 58.2 | 79.8 | 53.2 | 60.3 |
| Mistral CodeAlpaca | 7B | 35.4 | 50.2 | 34.6 | 8.3 | 56.1 | 79.1 | 54.2 | 60.9 |
| GLAN | 7B | 48.8 | **57.6** | 80.8 | 32.7 | **60.7** | **90.7** | **81.1** | **62.9** |

**Inference Cost** The inference cost of GLAN is closely tied to the models used for data generation. Note that GLAN is not limited to `GPT-4` or `GPT-3.5`; it can be applied to any open-source or closed-source models. To best showcase GLAN's performance, we selected the top available models at the time of writing—specifically, `GPT-4` and `GPT-3.5`. We queried `GPT-4` approximately 26,000 times for taxonomy, subject, and syllabus generation combined. For instruction generation, we queried `GPT-4` 10 million times, and for answer generation, we queried `GPT-3.5` also 10 million times. For more details, please refer to Appendix A.5.

## 3.2 MODEL TRAINING

We employ Mistral 7B (Jiang et al., 2023) as our base model. During training, we concatenate each instruction and response pair to a single sequence and only compute loss on response tokens. We train our model for 3 epochs with a learning rate of $3e$-6. The batch size is set to approximately 512 instruction-response pairs. We employ a dynamic batch size to ensure a constant total number of tokens per batch. We use a cosine learning rate schedule and we start with a linear warm-up of 1000 steps and the final learning rate is reduced to 0. The training requires approximately 8 days using 32 A100 GPUs.

## 3.3 BENCHMARK EVALUATION

The instruction data GLAN generated spans a wide range of subjects. We evaluate its effectiveness in mathematical reasoning, coding, logical reasoning, and academic exams.

***Mathematical Reasoning***: Mathematics is a common subject in many different disciplines. Hence, it is necessary to test the math reasoning ability of GLAN. We choose the two popular benchmarks for evaluation (i.e., GSM8K (Cobbe et al., 2021) and MATH (Hendrycks et al., 2021b)). GSM8K (Cobbe et al., 2021) is a high-quality math problem dataset that measures the basic multistep mathematical reasoning ability. It contains around 7k problems for training and 1K problems for test. MATH (Hendrycks et al., 2021b) is a challenging math dataset that contains mathematics competition-level problems from AMC, AIME, etc. The 7.5k training and 5K test problems cover seven math subjects, i.e., Prealgebra, Precalculus, Algebra, Intermediate Algebra, Number Theory, Counting and Probability, and Geometry. Note that GLAN does not use any examples in the training set of GSM8K or MATH. Following Luo et al. (2023a), we report 0-shot setting results for GLAN. ***Coding***: To evaluate the coding capability of GLAN, we opt for two coding benchmarks HumanEval (Chen et al., 2021) and MBPP (Austin et al., 2021). We employ 0-shot setting for HumanEval and 3-shot setting for MBPP following prior art (Chen et al., 2021; Luo et al., 2023b). ***BBH***: The instruction dataset we generated covers many disciplines, which can potentially enhance the reasoning ability of GLAN. Therefore, we evaluate GLAN on the BIG-Bench Hard dataset (BBH (Suzgun et al., 2023)), which contains 23 challenging tasks from Big-Bench (Srivastava et al., 2023). We employ the standard 3-shot setting with chain-of-thought demonstrations. ***Academic***

Table 2: Detailed Results on Academic Exam benchmarks.

| Model | ARC-E | ARC-C | MMLU | | | |
| --- | --- | --- | --- | --- | --- | --- |
| | | | STEM | Humanities | Social Sciences | Other |
| Mistral | 79.5 | 53.9 | 52.0 | 56.5 | 73.3 | 70.1 |
| GLAN | **90.7** | **81.1** | **60.1** | 54.9 | 71.8 | 68.6 |

***Exams***: We also evaluate GLAN on different academic benchmarks to verify whether GLAN is capable of solving exam questions. We choose two benchmarks (i.e., ARC (Clark et al., 2018) and MMLU (Hendrycks et al., 2021a)). Both benchmarks are composed of multi-choice questions. AI2 Reasoning Challenge (ARC (Clark et al., 2018)) contains grade-school level, multi-choice science questions. It contains two sub-sets, which are ARC-Challenge (ARC-C) and ARC-Easy (ARC-E). Massive Multitask Language Understanding (MMLU (Hendrycks et al., 2021a)) consists of a set of multiple-choice questions about 57 subjects ranging in difficulty from elementary levels to professional levels. It covers various of domains of knowledge, including humanities, STEM and social sciences. Note that there is a training set for ARC. However, we have excluded it from our training set during the decontamination process described in Section 3.1. Previous models mostly leverage probability-based methods on ARC and MMLU, which returns the best option based on the probabilities of the four options conditioned on the corresponding multi-choice question. We observe that after training on 10 million instructions, GLAN is able to *generate* its predicted options and analysis of multi-choice questions in plain text as `GPT-3.5` does. We therefore opt for 0-shot setting for GLAN and extract predictions using rules based on its completions as in Mitra et al. (2023).

**Results**   Our main results are shown in Table 1. We compare GLAN against general domain models (Orca 2 (Mitra et al., 2023), Mistral Instruct (Jiang et al., 2023) and WizardLM (Xu et al., 2023a)), math optimized models (MetaMath (Yu et al., 2024) and WizardMath (Luo et al., 2023a)) and coding optimized models (CodeAlpaca (Chaudhary, 2023)). We also report results of base LLMs (i.e., LLaMA2 (Touvron et al., 2023) and Mistral (Jiang et al., 2023)) as references. GLAN either obtains the best results or results close to the best across all benchmarks. We observe that capabilities of math or coding optimized models increase on math or coding benchmarks while usually not others. After instruction tuning, GLAN excels on multiple dimensions from mathematical reasoning, coding, reasoning, and academic exams with a systematical data generation approach. Also note that our method does not use any task-specific training data such as training sets of GSM8K, MATH, or ARC as in Orca 2, MetaMath, and WizardMath, which indicates the general applicability of GLAN.

**A Closer Look at Academic Exams**   ARC and MMLU are all multi-choice based benchmarks on academic exams. However, we observe that improvements of GLAN over Mistral on ARC are much larger than these on MMLU (see Table 1). By grouping the 57 subjects in MMLU into four categories (i.e., STEM, Humanities, Social Sciences, and Other (business, health, misc.)), we observe GLAN wildly improves on STEM in MMLU while not in other categories (Table 2). This is consistent with recent findings that Chain-of-Thought (CoT) primarily aids in symbolic reasoning problems rather than other types of questions (Sprague et al., 2024). Also note that ARC is composed of high school science problems, which are also STEM questions. GLAN is good at STEM subjects may be because responses of our dataset are from `GPT-3.5-turbo`, which by default generates responses with CoT reasoning. Indeed, we observe that GLAN generates solutions with CoT for multi-choice questions.

## 3.4   SCALING PROPERTY OF GLAN

We investigate the scaling property of GLAN by training Mistral on different numbers of examples (i.e., 50K, 200K, 500K, 1M, and 10M) we generated. The results on downstream tasks are shown in Figure 2. It can be observed that overall task performance tends to increase as we increase the data size. It's important to note the performance drop observed in the 200K to 1M data range for both HumanEval and BBH benchmarks. This regression might be attributed to the relatively small average number of data points per discipline at these scales. Our dataset encompasses 126 disciplines, with

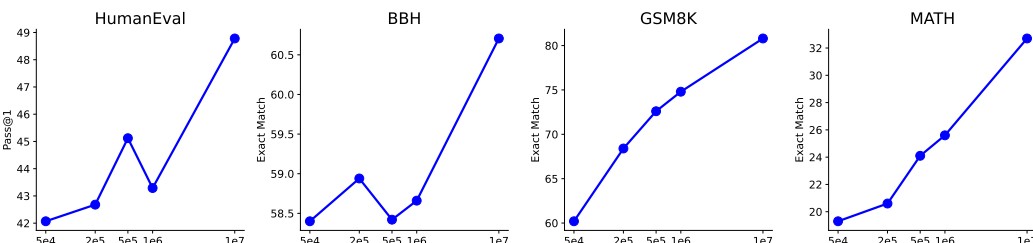

Figure 2: The scaling curve of GLAN on downstream tasks. The $x$-axis denotes GLAN data size (in $\log_{10}$ scale following (Kaplan et al., 2020)), and the $y$-axis denotes the task performance.

Table 3: The evaluation of loss values between the test data and training data. Large positive $\Delta$ (or $\Delta(\%)$) indicates task-specific in-domain training data might be exposed to the model during training.

| Benchmark/Loss | | LLaMA2-7B | Orca2-7B | Mistral-7B-Instruct | WizardLM-13B-V1.2 | GLAN-7B |
|---|---|---|---|---|---|---|
| **ARC-C** | $\Delta$ | -0.01 | 0.05 | -0.01 | -0.01 | -0.03 |
| | $\Delta(\%)$ | **-0.5%** | 2.10% | **-0.43%** | **-0.47%** | **-0.74%** |
| **ARC-E** | $\Delta$ | -0.02 | 0.04 | -0.03 | -0.02 | -0.01 |
| | $\Delta(\%)$ | **-0.95%** | 1.61% | **-1.19%** | **-0.91%** | **-0.23%** |
| **GSM8K** | $\Delta$ | 0 | 0.13 | 0 | 0.05 | 0.02 |
| | $\Delta(\%)$ | **0%** | 11.4% | **0%** | 4.39% | **0.92%** |
| **MATH** | $\Delta$ | -0.03 | 0.03 | -0.03 | -0.02 | -0.03 |
| | $\Delta(\%)$ | **-2.70%** | 2.54% | **-2.67%** | **-1.63%** | **-1.79%** |

an average of approximately 2,000 examples per discipline at the 200K total, increasing to about 8,000 examples per discipline at the 1M total. Interestingly, we observe a significant performance boost when scaling from 1M to 10M examples on both HumanEval and BBH. This improvement suggests that the increase in data points per domain crosses a threshold where it becomes substantial enough to positively impact model performance. Note that none of the curves have reached a plateau, indicating the potential for further improvement through continued scaling of GLAN. We leave further exploration on the scaling property of GLAN to future work.

## 3.5 TASK-SPECIFIC TRAINING DATA

GLAN is a generalized method to create synthetic data for instruction tuning. In order to evaluate the generalization capabilities of this synthetic data, we deliberately exclude task-specific training sets from all benchmarks on which we conduct our assessments. Similar to Wei et al. (2023), we explore whether models have been trained on task-specific in-domain data. We compute the training loss $L_{train}$ and test loss $L_{test}$ on ARC Challenge (ARC-C), GSM8K, and MATH for GLAN and other models in comparison. We choose these datasets because among all benchmarks evaluated in Section 3.3, these benchmarks contain training sets. Intuitively, the larger $\Delta = L_{test} - L_{train}$ is, the more likely the training set is exposed. To make $\Delta$ easier to interpret, we additionally compute the relative difference $\Delta(\%) = (L_{test} - L_{train})/L_{test}$. Table 3 shows the losses of the training and test splits for GLAN are nearly identical (or $\Delta$ is negative). This suggests that GLAN has not been exposed to in-domain data during training and tuning procedures. Please refer to the detailed losses of $L_{train}$ and $L_{test}$ in Table 8 (in Appendix). Additionally, as shown in Table 8, we observe that GLAN obtains higher losses on both test and training splits on GSM8K, MATH, and ARC compared to other models, while performances of GLAN on these datasets are high (see Table 1). This might imply that synthetic data generated by GLAN is diverse and our resulting model avoids convergence to any specific domain or style present in existing benchmarks.

## 3.6 INSTRUCTION FOLLOWING EVALUATION

**IFEval** We assess the instruction-following capabilities of GLAN utilizing the Instruction Following Evaluation dataset (IFEval (Zhou et al., 2023b)). IFEval consists of a collection of "verifiable instructions", encompassing 25 distinct types of instructions (around 500 prompts in total). Each prompt comprises one or more verifiable instructions. The evaluation involves four types

Table 4: Instruction following capability evaluation on IFEval.

| Model | Prompt-level strict-accuracy | Instruction-level strict-accuracy | Prompt-level strict-accuracy | Instruction-level loose-accuracy |
|---|---|---|---|---|
| GPT-3.5-turbo | 53.8 | 64.7 | 56.6 | 67.5 |
| GPT-4 | 77.1 | 83.7 | 79.7 | 85.6 |
| LLaMA2-7B | 14.8 | 27.1 | 16.6 | 29.4 |
| Orca2-7B | 19.4 | 28.9 | 26.1 | 34.7 |
| Mistral-7B-Instruct-v0.1 | 32.0 | 42.8 | 37.7 | 48.0 |
| WizardLM-13B-V1.2 | 23.1 | 33.5 | 26.6 | 37.6 |
| GLAN-7B | **34.0** | **44.8** | **41.2** | **51.6** |

of metrics at both prompt level and instruction level, evaluating strict and loose accuracies. As shown in Table 4, GLAN demonstrates superior instruction-following capabilities in both prompt-level and instruction-level evaluations. However, there is still a considerable gap compared to `GPT-3.5-turbo` and `GPT-4`.

**Evol-Instruct Test** Evol-Instruct testset (Xu et al., 2023a) contains real-world human instructions from diverse sources, and it consists of 218 instances with 29 distinct skills. Each instruction is associated with a difficulty level from 1 to 10. The responses are often open-ended descriptions, and we believe this benchmark is a necessary supplement to IFEval (answers to their instructions are "verifiable"). Following Xu et al. (2023a) and Chiang et al. (2023), we adopt a GPT-4-based automatic evaluation method to conduct a pairwise comparison between GLAN and other models. Specifically, GPT-4 is instructed to assign a score between 1 and 10 overall score w.r.t. the helpfulness, relevance, accuracy, and level of detail of responses generated by two different models for a given input question. A higher score indicates better overall performance. To mitigate potential order bias, we perform bidirectional comparisons for each response pair and determine their average score. The average score difference to GLAN (i.e., $\text{avg\_score}(\text{GLAN}) - \text{avg\_score}(x)$) serves as the final metric. Table 5 presents the results of pairwise comparisons across various levels of instruction difficulty. GLAN showcases superior performance compared to LLaMA-2, Orca 2, Mistral Instruct, and even WizardLM-13B (note that GLAN contains only 7B parameters) on most difficulty levels and overall scores. This suggests that GLAN demonstrates improved ability to process diverse instructions, regardless of their difficulty or complexity. Also, note that GLAN falls behind `GPT-3.5-turbo` as other models in comparison. Additionally, we group Evol-Instruct test according to the 29 skills and observe the same trends. Detailed results are listed in Appendix (Table 9 and 10). GLAN demonstrates strong performance on most skills, especially in Math, Coding, and Reasoning. However, it slightly falls short in common-sense related tasks. We also created GLAN-Test, similar to the Evol-Instruct Test but much larger in size, where GLAN outperforms other models as well (see Appendix A.9).

Table 5: Pairwise comparison on various difficulty levels between GLAN and other models on Evol-Instruct testset. The scores are the average gap of scores assigned by GPT-4, calculated as $\text{avg\_score}(\text{GLAN}) - \text{avg\_score}(x)$.

| Difficulty | Ratio | LLaMA2-7B | Orca2-7B | Mistral-7B-Instruct | Wizard-13B-V1.2 | GPT-3.5-turbo |
|---|---|---|---|---|---|---|
| (1-5) Easy | 41.00% | **5.46** | **2.19** | **1.13** | **1.32** | -1.22 |
| (6-10) Hard | 59.00% | **5.38** | **2.28** | **1.68** | **0.99** | -0.68 |

## 4 RELATED WORK

Recent literature has extensively explored the collection of various human-made resources for instruction tuning. An intuitive direction is to collect existing NLP datasets and corresponding task descriptions (Sanh et al., 2022; Wang et al., 2022; Zhou et al., 2023a), typical LLMs such as BLOOMZ (Muennighoff et al., 2023) and FLAN (Wei et al., 2022) are trained on this type of instruction tuning data. However, with only tens to thousands of existing datasets available, the scope and diversity of instruction tuning are inevitably limited. Another common practice is to implement instruction tuning with real-world human user prompts. For instance, InstructGPT (Ouyang et al., 2022) was trained on high-quality human prompts submitted by real-world users to OpenAI GPT

APIs. Vicuna (Chiang et al., 2023) leverages user-shared prompts along with ChatGPT responses for instruction tuning, and Dolly(Conover et al., 2023) was trained on simulated human-user interactions written by over 5k employees. Nevertheless, acquiring instructional data from human users typically involves high costs and involves privacy concerns.

As LLM capabilities improve, instruction tuning with LLM-generated data exhibits better scalability and potential in addressing the super-alignment problem (Shen et al., 2023). Leveraging the in-context learning ability of LLMs, Unnatural instructions (Honovich et al., 2023) and Self-instruct (Wang et al., 2023) sampled seed instructions as examples to elicit LLMs to generate new instructions. Taking advantage of the rephrasing ability of LLMs, WizardLM (Xu et al., 2023a) and WizardMath (Luo et al., 2023a) were trained using Evol-Instruct. Evol-Instruct iteratively employs ChatGPT to rewrite seed instructions into increasingly complex instructions. Similar to generation from seed instructions, carefully selected seed topics are used for generating textbook-like synthetic data (Li et al., 2023) or self-chat multi-turn dialogues (Xu et al., 2023b; Ding et al., 2023) for instruction tuning. However, models trained on these LLM-generated data only work well in specific domains such as math (Luo et al., 2023a; Yu et al., 2024), dialogue (Xu et al., 2023b; Ding et al., 2023) or open-ended question answering (Taori et al., 2023; Xu et al., 2023a). These methods encounter challenges in generalization (Gudibande et al., 2024), as the data diversity is restricted by seed instructions or seed topics.

## 5 CONCLUSIONS

We propose GLAN, a general and scalable method for synthesizing instruction data. Experiments show that GLAN can help large language models improve their capabilities in multiple dimensions, from mathematical reasoning, coding, academic exams, and logical reasoning to general instruction following. Currently, our synthetic data are based on the taxonomy of human knowledge and capabilities, and there are other types of useful data that have not been covered. We are interested in designing methods with border coverage. Our current instruction data are mostly question-answer pairs, and in the next step, we plan to generate synthetic data of multi-turn conversations and long documents.

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

# A APPENDIX

## A.1 LIMITATIONS

While GLAN presents significant advancements in academic benchmarks. However, there may still have several limitations in real world deployment. The resulting LLMs train on generated data using GLAN may occasionally produce factual incorrect (or even toxic) responses. Further training for refusal, hallucination reduction as well as toxic content reduction should be performed before deployment.

## A.2 BROADER IMPACTS

Data synthesizing is crucial for the continual scaling of large language models, especially as we exhaust available human data. GLAN demonstrates the potential to generate vast amounts of synthetic data from scratch, paving the way for even larger-scale data synthesis efforts. While GLAN has shown the effectiveness of synthetic data, we must point out that synthetic data may inherit and even amplify social biases present in the frontier LLMs for generation. Future research should focus on developing techniques to identify and correct biases in the generated datasets and models trained on them.

## A.3 PROMPT FOR SYLLABUS GENERATOR

The prompt template for syllabus generation is in Table 6.

Table 6: Prompt template for Syllabus Generator.

You are an expert in {`s.name`}.

Using the given data, design a syllabus for teaching students at the specified level.
Note that example subtopics or descriptions are just give you an impression of what this class like.
Feel free to add extra subtopics if needed (remember you are the expert in {`s.name`}).

Data:
- Level: {`s.level`}
- Main Topic: {`s.name`}
- Description or Example Subtopics: {`s.subtopics`}

### Syllabus Design Guide
1. **Introduction**: Start with an overview of the primary topic for the syllabus.
2. **Class Details**: For each class session, provide:
   - **Description**: Briefly describe the focus of the session.
   - **Knowledge Points**: Enumerate key concepts or topics.
   These will be used to craft homework questions.
   - **Learning Outcomes & Activities**: Offer expected learning results and suggest related exercises or activities.

## A.4 PROMPT FOR INSTRUCTION GENERATOR

The prompt template for instruction generator is in Table 7.

## A.5 DETAILED INFERENCE COST

In this paper, we pair GLAN with the closed-source models `GPT-4` and `GPT-3.5`. Since the architectures of these models are not publicly disclosed, we report API costs instead of actual computational costs (i.e., FLOPs). We estimate the API cost for generating 10 million data points to be approximately *360K USD* when using `GPT-4` and `GPT-3.5` for answer generation.

At the time of submission, we recommend using `GPT-4o` and `GPT-4o-mini` (for answer generation), reducing the cost to about *66K USD*. This is based on the consistent performance of `GPT-4o` over `GPT-4` and `GPT-4o-mini` over `GPT-3.5`. Additionally, leveraging `Mistral Large 2` and `Mistral 8x7B` (for answer generation) can further reduce costs to around *42K USD*.

Notably, API costs have significantly decreased over the past year, from *30/60 USD per million input/output tokens* to *2.5/10 USD per million input/output tokens*. We anticipate that these costs will continue to decline.

Moreover, open-source models, such as LLaMA-3 (GenAI, 2024), present powerful alternatives. The inference cost of GLAN when paired with these open-source models can be further reduced, making the application of GLAN more feasible.

Table 7: Prompt template for Instruction Generator.

```
## Background
- You are an expert in {s.name} education and you have designed a syllabus (i.e., '## Syllabus')
- We invite you (again) to design ONE homework question for given class sessions and some
knowledge points.
- The student have already learned all class sessions up to the current sessions
(i.e., '## Current Session(s)').
- There might be multiple class session in '## Current Session(s)'
- The designed homework question should focus on the topics in '## Current Session(s)' and you should
try to cover the given knowledge points in '## Given Knowledge Points'
- We prefer homework questions leveraging multiple knowledge points and across different topics

## Syllabus
{𝒜}

## Current Session(s)
{Ĉ}

## Given Knowledge Points
{𝕂̂}
```

## A.6 TASK-SPECIFIC TRAINING DATA

We provide the specific train/test values of different models on different benchmarks in Table 8.

Table 8: The evaluation of loss values between the test data and training data. Large positive $\Delta$ (or $\Delta(\%)$) indicate task specific in-domain training data may be exposed to the model during training.

| Benchmark/Loss | | LLaMA2-7B | Orca2-7B | Mistral-7B-Instruct | WizardLM-13B-V1.2 | GLAN-7B |
|---|---|---|---|---|---|---|
| **ARC-C** | $L_{test}$ | 2.02 | 2.39 | 2.32 | 2.11 | 4.03 |
| | $L_{train}$ | 2.03 | 2.34 | 2.33 | 2.12 | 4.06 |
| | $\Delta$ | -0.01 | 0.05 | -0.01 | -0.01 | -0.03 |
| | $\Delta\,(\%)$ | **-0.5%** | 2.10% | **-0.43%** | **-0.47%** | **-0.74%** |
| **ARC-E** | $L_{test}$ | 2.10 | 2.47 | 2.51 | 2.18 | 4.31 |
| | $L_{train}$ | 2.12 | 2.43 | 2.54 | 2.20 | 4.32 |
| | $\Delta$ | -0.02 | 0.04 | -0.03 | -0.02 | -0.01 |
| | $\Delta\,(\%)$ | **-0.95%** | 1.61% | **-1.19%** | **-0.91%** | **-0.23%** |
| **GSM8K** | $L_{test}$ | 1.38 | 1.14 | 1.26 | 1.14 | 2.17 |
| | $L_{train}$ | 1.38 | 1.01 | 1.26 | 1.09 | 2.15 |
| | $\Delta$ | 0 | 0.13 | 0 | 0.05 | 0.02 |
| | $\Delta\,(\%)$ | **0%** | 11.4% | **0%** | 4.39% | **0.92%** |
| **MATH** | $L_{test}$ | 1.11 | 1.18 | 1.12 | 1.22 | 1.67 |
| | $L_{train}$ | 1.14 | 1.15 | 1.15 | 1.24 | 1.70 |
| | $\Delta$ | -0.03 | 0.03 | -0.03 | -0.02 | -0.03 |
| | $\Delta\,(\%)$ | **-2.70%** | 2.54% | **-2.67%** | **-1.63%** | **-1.79%** |

## A.7 EVOL-INSTRUCT TEST RESULTS ON DIFFERENT DIFFICULTY LEVELS

The concrete Evol-Instruct test results on different difficulty levels are shown in Table 9.

## A.8 EVOL-INSTRUCT TEST RESULTS ON DIFFERENT SKILLS

The concrete Evol-Instruct test results on different skills are shown in Table 10.

## A.9 GLAN-TEST OVERALL RESULTS

**GLAN-Test** There are only hundreds of instructions in In IFEval and Evol-Instruct Test and we believe the domains or skills they can cover are rather limited. Therefore, we propose a held-out test set using GLAN data and we call it GLAN-Test. It contains 6,300 instructions on 126

Table 9: Pairwise comparison on various difficulty levels between GLAN and other models on Evol-Instruct testset. The scores are the average gap of scores assigned by GPT-4, calculated as `avg_score(GLAN) − avg_score(x)`.

| Difficulty | Ratio | LLaMA2-7B | Orca2-7B | Mistral-7B-Instruct | Wizard-13B-V1.2 | GPT-3.5-turbo |
|---|---|---|---|---|---|---|
| 1 | 5.1% | 5.41 | 2.23 | -0.37 | -0.21 | -2.41 |
| 2 | 8.7% | 5.87 | 1.74 | 1.06 | 1.41 | -1.18 |
| 3 | 12.4% | 5.72 | 2.35 | 1.04 | 1.37 | -1.14 |
| 4 | 10.5% | 5.61 | 1.34 | 1.52 | 1.54 | -0.92 |
| 5 | 4.1% | 4.67 | 3.31 | 2.39 | 2.5 | -0.45 |
| 6 | 19.3% | 4.43 | 2.42 | 0.74 | 1.54 | -1.36 |
| 7 | 11.0% | 4.97 | 1.26 | 1.62 | 1.36 | -0.41 |
| 8 | 17.9% | 6.02 | 3.58 | 3.17 | 1.7 | 0.15 |
| 9 | 6.0% | 6.35 | 4.2 | 1.36 | 0.9 | -0.92 |
| 10 | 5.1% | 5.14 | -0.05 | 1.53 | -0.54 | -0.85 |
| (1-5) Easy | 41.00% | **5.46** | **2.19** | **1.13** | **1.32** | -1.22 |
| (6-10) Hard | 59.00% | **5.38** | **2.28** | **1.68** | **0.99** | -0.68 |

Table 10: Pairwise comparison on various skills between GLAN and other models on Evol-Instruct testset. The scores are the average gap of scores assigned by GPT-4, calculated as `avg_score(GLAN) − avg_score(x)`.

| Skill | Ratio | LLaMA2-7B | Orca2-7B | Mistral-7B-Instruct | Wizard-13B-V1.2 | GPT-3.5-turbo |
|---|---|---|---|---|---|---|
| Math | 8.7% | 6.58 | 2.16 | 2.41 | 2.46 | -1.42 |
| Code Generation | 8.3% | 6.16 | 3.87 | 4.22 | 2.59 | -0.25 |
| Writting | 8.3% | 5.2 | 0.79 | -0.22 | 0.24 | -1.1 |
| Computer Science | 6.9% | 7.1 | 4.4 | 0.83 | 1.22 | 0.02 |
| Reasoning | 6.0% | 6.3 | 2.52 | 3.38 | 3.02 | 0.62 |
| Complex Format | 5.5% | 3.13 | 3.5 | -0.17 | 2.41 | -1.96 |
| Code Debug | 4.6% | 5.85 | 2.3 | 1.4 | 0.2 | -2.5 |
| Common-Sense | 4.1% | 6.5 | 3.19 | -1.33 | -0.92 | -2.78 |
| Counterfactual | 3.7% | 7.06 | 2.15 | 3 | 1.5 | 0.72 |
| Multilingual | 3.2% | 7.35 | 0.79 | 1.71 | -0.68 | -2.75 |
| Roleplay | 2.8% | 7.08 | 2.25 | 3.5 | 0.92 | -0.59 |
| Biology | 2.8% | 6.66 | 2.75 | 1.46 | -0.09 | 1.38 |
| Technology | 2.8% | -0.08 | 2.54 | -3 | -1.5 | -2.75 |
| Ethics | 2.8% | 6.59 | 3.38 | 2.41 | 5.42 | -0.21 |
| TruthfulQA | 2.3% | 3.1 | 3.7 | -1.05 | -1.3 | -0.85 |
| Sport | 2.3% | 4.3 | 0.55 | -0.2 | 4.8 | -0.3 |
| Law | 2.3% | 7.7 | 4.65 | 5.85 | 1.7 | 0.2 |
| Medicine | 2.3% | 3.9 | -2.05 | 1.9 | 0.15 | -1.25 |
| Literature | 2.3% | 6.3 | 1.9 | 0.2 | 1.45 | -0.15 |
| Entertainment | 2.3% | 4.5 | 2.7 | -3 | 1.9 | -3.2 |
| Art | 2.3% | 4.9 | 1 | 2.9 | -0.85 | -2.05 |
| Music | 2.3% | 4.4 | 4.1 | 0.5 | 1.45 | -2.3 |
| Toxicity | 1.8% | 7.25 | 3.12 | 3.75 | 1.63 | -1.32 |
| Economy | 2.3% | 6 | 0.15 | 1.9 | 0 | 0 |
| Physics | 2.3% | 6.8 | 2.5 | 4.35 | 3.65 | -1 |
| History | 1.8% | 4.12 | -0.56 | 3.76 | -0.31 | 0.12 |
| Academic Writing | 1.8% | 6.76 | 6.37 | 2.44 | 1.37 | 0.62 |
| Chemistry | 0.9% | 9.5 | 0.63 | 5.25 | 2.5 | 0.75 |
| Philosophy | 0.5% | 11 | -0.25 | 0.25 | -0.25 | 0.5 |
| Avg.(29 skills) | 100% | 5.42 | 2.24 | 1.41 | 1.16 | -0.95 |

disciplines (50 instructions for each discipline). We further categorize the 126 disciplines to 8 distinct *fields* (i.e., Academic-Humanities, Academic-Social Science, Academic-Natural Science, Academic-Applied Science, Academic-Formal Science, Industry-Manufacturing, Industry-Services and Industry-Agriculture). We believe that the extensive domain coverage of GLAN-Test renders it an effective test bed for the assessment of generalization capabilities in LLMs. We adopt the same GPT-4 based evaluation protocol as in Evol-Instruct Test (previous paragraph). We prompt GPT-4 to do a pairwise ranking of GLAN and other models in comparison. The overall results and results across the 8 fields are presented in Table 11, where GLAN obtains higher GPT-4 scores than Orca2-7B, Mistral-7B Instruct and WizardLM-13B, despite using only 7B parameters. GLAN still lag behind `GPT-4`. Detailed results for the 126 fine-grained disciplines can be found in Appendix A.10 (see Table 12 for more details). GLAN demonstrates its effectiveness on multiple domains (or

disciplines) such as Mathematics, Physics, Chemistry, Computer science, Electrical, Mechanical, etc., indicating that smaller models may yield general improvements on various domains through strategic fine-tuning. Furthermore, it is noted that GLAN demonstrates less-than-ideal performance across distinct disciplines such as American history, Divinity, or Radiology. This observation underscores the potential for further refinement and development of our methodology within these domains.

Table 11: Pairwise comparison between GLAN and other models on GLAN-Test (the 126 disciplines are categorized into 8 fields for clarity of the illustration). The scores are the average gap of scores assigned by GPT-4, calculated as `avg_score(GLAN)` − `avg_score(x)`.

| Field (Ratio) | Orca2-7B | Mistral-7B-Instruct | WizardLM-13B-V1.2 | GPT-4 |
|---|---|---|---|---|
| Academic-Humanities (15.9%) | 0.79 | 0.25 | 0.02 | -0.62 |
| Academic-Social Science (7.9%) | 1.22 | 0.21 | 0.09 | -0.63 |
| Academic-Natural Science (4.0%) | 1.73 | 1.23 | 0.53 | -0.5 |
| Academic-Applied Science (42.1%) | 1.58 | 0.32 | 0.08 | -0.58 |
| Academic-Formal Science (3.2%) | 3.87 | 2.48 | 2.32 | -0.55 |
| Industry-Manufacturing (12.7%) | 2.26 | 0.56 | 0.33 | -0.43 |
| Industry-Services (11.9%) | 1.82 | 0.23 | 0.09 | -0.5 |
| Industry-Agriculture (2.4%) | 1.2 | 0.46 | 0.13 | -0.33 |
| Overall (100.0%) | **1.61** | **0.43** | **0.19** | -0.55 |

## A.10 GLAN-TEST RESULTS ON DIFFERENT DISCIPLINES

Table 12: Pairwise comparison across 126 disciplines (or domains) on *GLAN-Test*. The scores are generated from the average gap between GLAN and other model $x$ in assessment scores assigned by GPT-4, calculated as `avg_score(GLAN) − avg_score(x)`.

| Discipline | Orca-2-7b | Mistral-7B-Instruct-v0.1 | WizardLM-13B-V1.2 | GPT-4 |
|---|---|---|---|---|
| Avg. | 1.61 | 0.43 | 0.19 | -0.55 |
| Advertising | 1.92 | 0.46 | 0.21 | -0.04 |
| Aerospace industry | 3.24 | 1.24 | 0.6 | -0.42 |
| Agriculture | 2.44 | 0.04 | -0.05 | -0.48 |
| American history | -0.49 | -0.27 | -0.76 | -0.83 |
| American politics | 1.23 | -0.3 | -0.4 | -0.87 |
| Anthropology | 0.59 | 0.17 | 0.06 | -0.27 |
| Applied mathematics | 3.75 | 2.6 | 2.74 | -0.47 |
| Archaeology | 2.59 | -0.11 | 0.1 | -0.56 |
| Architecture and design | 2.63 | 0.34 | 0.4 | -0.37 |
| Astronomy | 1.01 | 0.83 | 0.03 | -0.44 |
| Automotive industry | 1.27 | 0.71 | 0.46 | -0.06 |
| Biblical studies | -0.05 | 0.33 | -0.47 | -0.65 |
| Biology | 1.09 | 0.22 | -0.09 | -0.17 |
| Business | 3.61 | 1.14 | 0.88 | -0.26 |
| Chemical Engineering | 3.15 | 1.6 | 1.18 | -0.77 |
| Chemistry | 3.06 | 2.09 | 0.8 | -0.87 |
| Civil Engineering | 1.94 | 0.74 | 0.75 | -0.25 |
| Clinical laboratory sciences | 1.32 | 0.94 | -0.11 | -0.47 |
| Clinical neuropsychology | 2.15 | 0.29 | 0.25 | -0.4 |
| Clinical physiology | 2.07 | 0.41 | 0.51 | -0.08 |
| Communication studies | 0.3 | 0.26 | -0.15 | -0.3 |
| Computer science | 4.29 | 1.45 | 1.9 | -0.33 |
| Cultural industry | 3.15 | 0.44 | 0.05 | -0.36 |
| Dance | 2.11 | 0.21 | 0.4 | -0.47 |
| Dentistry | 1.67 | 0.66 | 0.48 | 0.01 |
| Dermatology | 2.12 | 0.55 | -0.05 | -0.65 |
| Divinity | -0.34 | -0.17 | -0.48 | -0.89 |
| Earth science | 0.39 | 0.44 | -0.08 | -0.33 |
| Economics | 2.62 | 0.96 | 0.62 | -0.4 |
| Education | 2.67 | 0.42 | 0.2 | -0.84 |
| Education industry | 2.19 | 0.4 | 0.56 | -1.33 |
| Electric power industry | 3.23 | 1.31 | 0.39 | -0.79 |
| Electrical Engineering | 3.81 | 1.26 | 1.41 | -0.34 |
| Emergency medicine | 2.04 | 0.44 | -0.18 | -0.86 |
| Energy industry | 3.59 | 0.98 | 0.54 | -0.22 |
| Environmental studies and forestry | 0.12 | 0.41 | 0.1 | -0.45 |
| Epidemiology | 3.02 | 0.52 | 0.33 | -0.46 |
| European history | 0.14 | 0.62 | 0.15 | -0.18 |
| Fashion | 2.5 | 0.66 | 0.47 | -0.53 |
| Film | 0.76 | 0.45 | -0.16 | -0.78 |
| Film industry | 1.58 | 0.46 | 0.25 | -0.59 |
| Fishing industry | 1.67 | 1 | 0.57 | -0.09 |
| Floral | 1.92 | 0.89 | 0.58 | -0.09 |
| Food industry | 3.64 | 0.12 | 0.14 | -0.42 |
| Foreign policy | 2.4 | 0.49 | 0.16 | -0.46 |
| Geography | 0.88 | 0.6 | 0.28 | -0.66 |
| Geriatrics | 2.19 | -0.32 | -0.56 | -0.71 |
| Gynaecology | 1.05 | -0.27 | -0.26 | -0.67 |
| Healthcare industry | 1.62 | -0.25 | 0.14 | -0.5 |
| Hematology | 0.35 | 0.32 | -0.05 | -0.72 |
| History | 0.75 | 0.54 | -0.04 | -0.38 |
| Holistic medicine | 0.85 | 0.48 | 0.26 | -0.27 |
| Hospitality industry | 2.36 | 0.48 | 0.28 | -0.07 |
| Housing | 4.04 | 0.15 | -0.22 | -0.62 |
| Industrial robot industry | 3.84 | 1.22 | 0.84 | -0.71 |
| Infectious disease | 1.76 | 0.14 | 0.18 | -0.56 |
| Insurance industry | 2.67 | 0.42 | 0.61 | -0.4 |
| Intensive care medicine | 1.11 | 0.56 | 0.08 | -0.33 |
| Internal medicine | 1.02 | 0.45 | -0.01 | -0.42 |
| Journalism | 2.77 | -0.13 | -0.21 | -0.69 |
| Languages and literature | 0.45 | 0.05 | -0.39 | -0.84 |
| Law | 0.42 | 0.39 | 0.04 | -0.49 |
| Leisure industry | 1.49 | 0.12 | -0.09 | -0.49 |
| Library and museum studies | 1.52 | 0.5 | 0.33 | -0.32 |

| Discipline | Orca-2-7b | Mistral-7B-Instruct-v0.1 | WizardLM-13B-V1.2 | GPT-4 |
|---|---|---|---|---|
| Linguistics | 0.39 | 0.38 | -0.12 | -0.96 |
| Logic | 2.95 | 1.56 | 1.62 | -0.79 |
| Materials Science and Engineering | 1.71 | 0.97 | 0.54 | -0.91 |
| Mathematics | 4.69 | 3.81 | 2.73 | -0.61 |
| Mechanical Engineering | 2.25 | 1.71 | 1.15 | -0.95 |
| Medical toxicology | 0.62 | 0 | 0.11 | -1.01 |
| Medicine | 1.49 | 0.93 | 0.36 | -0.37 |
| Military sciences | 0.42 | 0.53 | 0.17 | -0.45 |
| Mining | 3.17 | 0.32 | 0.41 | -0.61 |
| Music | 2.85 | 0.38 | 1.07 | -0.05 |
| Music industry | 2.05 | -0.03 | -0.08 | -0.8 |
| Nursing | 1.49 | 0.14 | -0.12 | -0.59 |
| Nutrition | 1.15 | -0.2 | -0.13 | -0.65 |
| Obstetrics | 1.49 | 0.08 | -0.43 | -0.53 |
| Ophthalmology | 0.97 | 0.01 | -0.47 | -0.97 |
| Otolaryngology | 1.51 | -0.44 | -0.29 | -1.11 |
| Pathology | 0.23 | 0.35 | 0.19 | -0.72 |
| Pediatrics | 1.62 | 0.55 | -0.34 | -0.47 |
| Performing arts | 0.38 | 0.09 | -0.36 | -1.06 |
| Petroleum industry | 3.12 | 0.44 | 0.08 | -0.54 |
| Pharmaceutical industry | 2.75 | 0.41 | 0.4 | -0.46 |
| Pharmaceutical sciences | 0.77 | 0.19 | 0.16 | -0.8 |
| Philosophy | 0.51 | 0.25 | 0.49 | -0.64 |
| Physics | 3.15 | 2.67 | 2.05 | -0.73 |
| Political science | 0.04 | -0.05 | -0.31 | -0.91 |
| Prehistory | 0.35 | 0.19 | 0.05 | -0.41 |
| Preventive medicine | 2.69 | 0.57 | 0.09 | -0.36 |
| Psychiatry | 2.93 | 0.27 | -0.07 | -0.32 |
| Psychology | 0.53 | -0.02 | -0.3 | -0.96 |
| Public administration | 0.94 | -0.27 | 0.1 | -1.2 |
| Public health | 1.21 | 0.07 | 0.22 | -0.56 |
| Public policy | 0.78 | -0.06 | -0.28 | -0.92 |
| Pulp and paper industry | 1.13 | 0.63 | 0.57 | -0.25 |
| Radiology | -0.17 | -0.19 | -0.82 | -0.62 |
| Real estate industry | 1.01 | 0.02 | -0.12 | -0.5 |
| Religious Studies | 0.38 | 0 | -0.32 | -0.63 |
| Retail industry | 1.1 | -0.25 | -0.37 | -0.6 |
| Semiconductor industry | 1.49 | 0.64 | 0.71 | -0.42 |
| Sexology | 1.81 | -0.44 | -0.37 | -0.96 |
| Shipbuilding industry | 1.54 | 0.37 | 0.42 | -0.32 |
| Social work | 0.93 | -0.42 | -0.53 | -0.77 |
| Sociology | 1.49 | 0.21 | 0.76 | -0.3 |
| Steel industry | 0.88 | 0.45 | 0.09 | -0.34 |
| Surgery | 0.86 | -0.02 | -0.35 | -0.73 |
| Systems science | 1.9 | 0.56 | 0.41 | -0.45 |
| Telecommunications industry | 1.81 | 0.4 | 0.39 | -0.27 |
| Television | 0.37 | -0.33 | -0.69 | -1 |
| Textile industry | 0.82 | -0.26 | -0.68 | -0.59 |
| Theatre | 0.31 | -0.27 | -0.34 | -1.07 |
| Theology | -0.38 | 0.37 | -0.45 | -0.54 |
| Tobacco industry | 0.59 | -0.13 | -0.48 | -0.67 |
| Transport industry | 1.19 | -0.33 | -0.36 | -0.56 |
| Transportation | 1.74 | 0.26 | 0.17 | -0.74 |
| Urology | 0.05 | -0.29 | -0.36 | -0.64 |
| Veterinary medicine | -0.14 | 0.36 | -0.31 | -0.62 |
| Video game industry | 1.67 | 0.2 | -0.24 | -0.62 |
| Visual arts | 0.98 | 0.22 | 0.26 | -0.56 |
| Water industry | 0.9 | -0.11 | -0.09 | -0.51 |
| Wood industry | 1.36 | 0.5 | 0.31 | -0.25 |