# OpenReview forum: "Synthetic Data (Almost) from Scratch: Generalized Instruction Tuning for Language Models"
_ICLR.cc/2025/Conference — Submitted to ICLR 2025_

### Official Review · Reviewer_1yM3 · 2024-10-16

**Soundness:** 3
**Presentation:** 3
**Contribution:** 3
**Rating:** 6
**Confidence:** 4

**Summary:**

The introduces Generalized Instruction Tuning (GLAN), a novel method for large-scale instruction tuning of language models. Unlike previous methods that rely on seed examples or existing datasets, GLAN creates synthetic data using a pre-defined taxonomy of human knowledge. This taxonomy is derived from a structure similar to human education systems and is used to generate diverse instructions across various domains. GLAN's approach is scalable, customizable, and supports adding new fields by incorporating new nodes into the taxonomy. Extensive experiments demonstrate that GLAN improves performance across a range of tasks, including mathematical reasoning, coding, and general instruction following, without using task-specific training data.

**Strengths:**

Scalability and Coverage: GLAN generates synthetic instruction data across a broad range of domains using a hierarchical taxonomy of human knowledge, ensuring comprehensive coverage and scalability without relying on pre-existing datasets.

Customization and Flexibility: The method allows for easy integration of new fields or disciplines by adding nodes to the taxonomy, providing flexibility for expanding the instruction tuning process to emerging or niche areas.

Strong Empirical Performance: Extensive experiments show that GLAN significantly enhances model performance in tasks like mathematical reasoning, coding, and academic exams, outperforming or matching state-of-the-art models without task-specific data.

**Weaknesses:**

Method Feasibility: The method is somewhat novel, but its feasibility in the real world is questionable. While the comparative methods could generate instructions for self-training, I suspect that GLAN is bounded by the knowledge of the generator, e.g., GPT-4 in this paper. It significantly impedes the deployment of GLAN as a method to instruction-tune frontier models, if not to distill from GPT-4 with yet another trick.

Baselines: With the last point being said, at least a few comparative methods to generate instructions, like self-instruct or evol-instruct, should be compared with GLAN. After all, the paper claims the instruction generation method as its core contribution. However, no baseline method is even compared.

Lack of Evaluation on Bias and Ethical Concerns: The paper acknowledges the potential risks of amplifying biases in generated data but does not present specific strategies for bias detection or mitigation, which is critical for real-world deployment.

**Questions:**

- Table 1: how many instructions are there to train each model?
- Table 2: provide a more in-depth analysis of why GLAN can't even beat the base model on non-STEM MMLU subjects.
- Generate instructions from an open-source frontier model and train on it to test the self-training capabilities of GLAN.
- The scaling trends look really promising! But 1. how do GLAN instructions compare with other methods given the same quantity? can plot as a comparison; 2. will the performance ever saturate? worth more experiments and analysis.
- Does GLAN generate longer instructions than other methods? I think it might be a suitable method to scale the length of instructions, which might be an interesting direction.

---

> ### Author Response · Authors · 2024-11-24
> **Response to Reviewer 1yM3 (1/3)**
>
> > Method Feasibility: The method is somewhat novel, but its feasibility in the real world is questionable. While the comparative methods could generate instructions for self-training, I suspect that GLAN is bounded by the knowledge of the generator, e.g., GPT-4 in this paper. It significantly impedes the deployment of GLAN as a method to instruction-tune frontier models, if not to distill from GPT-4 with yet another trick.
>
> Thank you very much for your insightful comments, which are truly inspiring for future research in this direction. However, we would like to clarify several aspects.
> ## On “Comparative methods could generate instructions for self-training” and “to instruction-tune frontier models”
> First, we must clarify that "generating instructions for self-training" is not the same as "instruction-tuning frontier models." The only comparative method that has demonstrated the ability for self-training is self-instruct, which expands 175 seed instruction-response pairs using GPT-3 base (davinci) to generate additional instructions. These instructions are then used to fine-tune GPT-3 base. The resulting *fine-tuned* GPT-3 model (GPT-3 Self-Instruct) significantly outperforms GPT-3 base and is only slightly behind GPT-3 Instruct (text-davinci-001) on SuperNI evaluations.
> However, improving a frontier LLM (which is already fine-tuned through supervised training and further refined using RLHF or PPO) is considerably more challenging than improving a base model.
>
> ## On “feasibility in the real world is questionable”
> In many real-world applications, such as mobile or desktop software, deploying frontier LLMs locally is often impractical due to resource constraints. Small language models (SLMs), such as those with 3B or 7B parameters, offer a more cost-effective solution and are well-suited for local deployment. Our research demonstrates that GLAN can substantially enhance the performance and capabilities of these SLMs, making them a viable alternative for resource-limited scenarios.
>
> ## On instruction-tune frontier models or improving frontier models
> To systematically improve frontier LLMs, a broad and diverse set of prompts is essential for identifying areas or domains that require improvement. GLAN can generate instructions across a wide range of domains and tasks, providing the fuels needed for self-improvement in frontier LLMs.
> Besides the initial input prompts, we believe a good reward model is also needed to guide the improvement of response quality.
>
> > Baselines: With the last point being said, at least a few comparative methods to generate instructions, like self-instruct or evol-instruct, should be compared with GLAN. After all, the paper claims the instruction generation method as its core contribution. However, no baseline method is even compared.
>
> Thank you for highlighting this! We have actually compared GLAN with both self-instruct and evol-instruct in Table 1. Specifically, “WizardLM v1.2” [1] is based on evol-instruct, while “Mistral CodeAlpaca” [2], which focuses on the coding domain, is based on self-instruct. GLAN outperforms both across all benchmarks.
> We will include these clarifications about WizardLM v1.2 and Mistral CodeAlpaca in the revised version of the paper.
>
>
> [1] Xu et al. (2024). WizardLM: Empowering Large Language Models to Follow Complex Instructions. https://arxiv.org/pdf/2304.12244
>
> [2] https://github.com/sahil280114/codealpaca

---

> ### Author Response · Authors · 2024-11-24
> **Response to Reviewer 1yM3 (2/3)**
>
> > Lack of Evaluation on Bias and Ethical Concerns: The paper acknowledges the potential risks of amplifying biases in generated data but does not present specific strategies for bias detection or mitigation, which is critical for real-world deployment.
>
> We sincerely appreciate your thoughtful feedback regarding bias and ethical concerns. These are undoubtedly critical considerations for the deployment of LLMs in real-world applications. While we acknowledge the importance of addressing these challenges, this work primarily focuses on *synthetic data generation*. As such, a comprehensive exploration of bias detection and mitigation strategies is beyond the immediate scope of this research.
>
> Nevertheless, to detect and mitigate bias and ethical issues in our method/model (or other models trained on synthetic instructional data), one can adopt methods for bias detection and mitigation in recent work [1][2][3].
>
> [1]  Fan, et al. (2024). BiasAlert: A Plug-and-play Tool for Social Bias Detection in LLMs. https://arxiv.org/abs/2407.10241
>
> [2] Narayan et al. (2024). Bias Neutralization Framework: Measuring Fairness in Large Language Models with Bias Intelligence Quotient (BiQ). https://arxiv.org/abs/2404.18276
>
> [3] Gallegos et al., (2024).  Bias and Fairness in Large Language Models: A Survey. https://aclanthology.org/2024.cl-3.8/ Computational Linguistics, Volume 50, Issue 3 - September 2024
>
>
>
> > Table 1: how many instructions are there to train each model?
>
> | Models             | \# instructions |
> | ------------------ | --------------- |
> | GPT-4              | unknown         |
> | GPT-3.5-turbo      | unknown         |
> | Orca 2             | 11.8M           |
> | WizardLM v1.2      | \>= 250K        |
> | Mistral Instruct   | unknown         |
> | MetaMath Mistral   | 395K            |
> | WizardMath v1.1    | unknown         |
> | Mistral CodeAlpaca | 20K             |
> | GLAN               | 10M             |
>
>
>
>
>
> As stated in [1], WizardLM v1.0 is trained on 250K instructions. The exact number of instructions used for WizardLM v1.2 is not disclosed, but we assume that at least the same quantity was used. Similarly, [2] discusses WizardMath v1.0 but does not specify the exact number of instructions used for training. It also remains unclear whether additional instructions were used for training WizardMath v1.2.
>
> We would also like to highlight that the 10M instructions we generated include tasks beyond the scope of current benchmarks, such as instructions related to fishing and textiles. To the best of our knowledge, none of the benchmarks used in this paper cover these domains. Therefore, solely evaluating GLAN on existing benchmarks may not provide a comprehensive assessment of its capabilities. This limitation on evaluations similarly applies to frontier LLMs.
>
> [1]. C. Xu et al. “WizardLM: Empowering Large Language Models to Follow Complex Instructions”. 2023
>
> [2] Luo et al. “WizardMath: Empowering Mathematical Reasoning for Large Language Models via Reinforced Evol-Instruct”

---

> ### Author Response · Authors · 2024-11-24
> **Response to Reviewer 1yM3 (3/3)**
>
> > Generate instructions from an open-source frontier model and train on it to test the self-training capabilities of GLAN.
>
> > The scaling trends look really promising! But 1. how do GLAN instructions compare with other methods given the same quantity? can plot as a comparison; 2. will the performance ever saturate? worth more experiments and analysis.
>
> We sincerely appreciate your constructive suggestions and agree that the proposed experiments would provide valuable insights into our approach. However, conducting these experiments requires significant computational resources and time to obtain reliable results. While we have already begun exploring this direction, the two-week rebuttal period is insufficient to produce meaningful findings. We intend to incorporate these results into future work to provide a more comprehensive analysis.
>
> > Does GLAN generate longer instructions than other methods? I think it might be a suitable method to scale the length of instructions, which might be an interesting direction.
>
> Thanks for your interesting question. To answer this question, we compute the average prompt and response length of GLAN-10M, Alpaca-52k (self-instruct) and WizardLM_evol_instruct_70k (evol-instruct). The results are as follows and as you expected:
>
> | method        | Average prompt length | Average response length |
> | ------------- | --------------------- | ----------------------- |
> | GLAN          | 128.07                | 444.52                  |
> | Self instruct | 61.61                 | 64.51                   |
> | Evol Instruct | 122.04                | 347.48                  |
>
> This may be due to GLAN's ability to generate complex instructions in areas such as math, physics problem-solving, essay writing and more.
>
> Thank you once again for your valuable feedback! We will incorporate all the results and explanations mentioned above into the revised manuscript.

---

> > ### Comment · Reviewer_1yM3 · 2024-11-25
> > **Thank you for your response**
> >
> > Thank you for your response. Most of my concerns are well addressed. I'd like to raise the rating.

---

> > > ### Author Response · Authors · 2024-11-26
> > >
> > > Thank you for the updated score! We're glad to hear that most of your concerns were addressed.

---

### Official Review · Reviewer_h7cZ · 2024-10-24

**Soundness:** 2
**Presentation:** 2
**Contribution:** 2
**Rating:** 5
**Confidence:** 5

**Summary:**

This paper introduces a Generalized Instruction Tuning (GLAN) framework for LLM's instruction tuning. The authors leverage GPT4 to build a knowledge taxonomy on various fields, and sequentially ask GPT4 to break down fields into subfields, disciplines, subjects, syllabus, class sessions, and key concepts. After that, they repeatedly prompt GPT4/GPT3.5 to generate instruction-tuning data (including questions and answers) according to randomly sampled class sessions and key concepts within a syllabus. Finally, 10M instruction-response pairs are generated to align a pre-trained LLM (e.g., Mistral) to a chatbot via instruction tuning.

**Strengths:**

1. Building the knowledge taxonomy and structure is an effective approach to ensuring diversity and knowledge coverage when generating training samples.
2. The curated 10M instruction-response pairs are valuable for the LLM community.

**Weaknesses:**

1. My core concern lies in the novelty.  Basically, GLAN is distilling GPT4 to generate instruction-response pairs, and the idea of constructing a knowledge structure has also been explored by previous works[1][2]. I hope the authors can clarify their research contribution.
2. The scalability of the proposed method is questionable. For the human verification of knowledge taxonomy, there seems 126 * 200 * 30 * 5 = 3M key concepts according to Section 3.1, and the human checking cost is definitely not 'limited' as claimed by the authors. For the data generation by GPT models, this paper queried GPT4/3.5 for 10M times and cost at least 42K USD, which is far more expensive than current data generation methods.
3. The presentation of the proposed knowledge taxonomy is confusing. It is recommended to add a figure or a table in the early pages of this paper, to illustrate the definition, examples, and final quantity of the knowledge hierarchy (fields, subfields, disciplines, subjects, syllabus, class sessions, and key concepts).
4. The experiments are insufficient. The method is only evaluated on the Mistral model.

[1] Camel: Communicative agents for" mind" exploration of large language model society. NeurIPS 2023.

[2] Mathscale: Scaling instruction tuning for mathematical reasoning. ICML 2024.

**Questions:**

1. According to Section 2.1, is the field-subfield-discipline hierarchy discarded after collecting the leaf nodes of the discipline list？
2. In Table1, why is the performance of Mistral-Instruct worse than the base model Mistral on MBPP, GSM8K, and MMLU?

**Details Of Ethics Concerns:**

The paper introduces a synthetic dataset, and the data discrimination, bias, and fairness should be examined.

---

> ### Author Response · Authors · 2024-11-24
> **Response to Reviewer h7cZ (Part 1/2)**
>
> We sincerely thank Reviewer h7cZ for your review and are grateful for the time you spent on our submission. We are glad for the acknowledgment that our approach is effective for ensuring diversity of instruction tuning and valuable for the LLM community. Below we would like to give detailed responses to each of your comments.
>
> **W1: novelty and research contribution of GLAN**
>
> We would like to highlight the research contribution of our paper to address potential concerns regarding novelty.
>
> - As far as we know, GLAN is the first method that **does not rely on existing seed topics or instructions to synthesize data from scratch**. This approach brings scalability and flexibility to instruction tuning data, especially by filling in the gaps for underrepresented and niche domains.
> - In terms of taxonomy construction, our method first induces LLMs to **recursively decompose high-level concepts** into smaller concepts for instruction generation by autonomously searching within the knowledge space of large LLMs. This approach offers unique advantages, including better coverage and high automation.
> - At the performance aspect, GLAN surpasses the progress seen across many open instruction model recipes released recently. Unlike previous instruction tuning methods that are limited to benefiting specific tasks, GLAN demonstrates **consistent improvements across a wide range of tasks and domains**.
> - At the **data and resource** aspect, as recognized, the curated **10M high quality instruction-response pairs** are valuable for future work.
>
>
> Compared to the mentioned works (Camel and MathScale), we would like to highlight the unique strengths of GLAN:
>
> * **Generalizability:** Unlike Camel, which focuses on role-playing tasks, and MathScale, which is tailored to the mathematics domain, GLAN aims to enhance capabilities across a broad range of tasks. GLAN covers a wide range of domains, with mathematics being just one of them.
> * **Scalability:** GLAN can generate over 500 million unique combinations of knowledge points for instructional data generation, making it highly scalable.
> * **Customizability:** New fields, domains, or disciplines can be easily added to GLAN's taxonomy, providing flexibility and adaptability.
>
> We will add an in-depth discussion on comparison with the mentioned works in the revised version.
>
> **W2: The scalability of the proposed method is questionable.**
> We hope to clarify that our method's scalability is not limited by human verification and API costs, the reasons are as follows:
>
> **A. Human Verification**
>
> - First, as mentioned in lines 153 and 89, human verification is **at the discipline level rather than at the concept level**. The cost of human verification is low due to the limited number of disciplines in the taxonomy (126 disciplines in total).
> - Second, human verification is **only required in the initial step** when constructing the taxonomy. Subsequent steps in the GLAN process are fully automated, significantly minimizing the cost of human verification. Since we have already produced the entire taxonomy, **subsequent scaling up no longer requires any additional human verification.**
>
> **B. LLM API Cost**
> We would like to clarify that the whole GLAN methodology for data synthesizing is not limited by API costs.
>
> It is a **model-agnostic methodology,** allowing users to select any open-source or closed-source models based on their requirements. For example, users who deploy free open-source models such as Llama-3 72B will incur a data generation cost of $0.
>
> In our main implementation, we adopted GPT4 as the large model for distillation only to obtain a higher quality of instruction tuning data for future work.
>
> **W3: the presentation of the proposed knowledge taxonomy**
> Thanks for your great suggestion. We'll add a figure in the revised version to illustrate the definition, examples, and final quantity of the knowledge hierarchy for better understanding.
>
> **W4: "The method is only evaluated on the Mistral model."**
> As shown by previous works \[1\], the effectiveness of general-purpose instruction-tuning data is **consistent across different models**. Our experiments follow a similar setting with similar works, such as Orca \[2\], Phi \[3\], and WizardLM \[4\]. When conducting our experiments, we utilized the Mistral-series models, which were among the best open-source models available. We believe that the evaluation results of the Mistral models are **transferable** to more recent open-source models.
>
> It is worth mentioning that GLAN surpasses the progress seen across various open instruction model recipes released recently. As shown in Table 1, the Mistral 7B model tuned with GLAN has already surpassed all other models of similar size in other open instruction model recipes. The GLAN-7B model even outperforms the 13B WizardLM v1.2 on most benchmarks.

---

> > ### Author Response · Authors · 2024-11-24
> > **Response to Reviewer h7cZ (Part 2/2)**
> >
> > **Q1: "According to Section 2.1, is the field-subfield-discipline hierarchy discarded after collecting the leaf nodes of the discipline list？"**
> > No, in addition to the information from the leaf nodes, the hierarchy information from field to subject is also utilized during the instruction generation phase.
> >
> > As shown in the prompt template for the instruction generator (Appendix A.4), we provide the complete Syllabus and Current Session to which the current knowledge point belongs as additional context to improve the quality of the generated instructions.
> >
> > **Q2: "In Table1, why is the performance of Mistral-Instruct worse than the base model Mistral on MBPP, GSM8K, and MMLU?"**
> > The main reason is that instruction-tuning, especially general-purpose instruction tuning, primarily aims to enhance the model's ability to follow user instructions, which may impair the model's performance on specific tasks \[1\]. Especially for tasks like MBPP, GSM8K, and MMLU, where most answers come from pre-trained knowledge, the model may learn to generate more human-friendly and less harmful text during instruction tuning, at the cost of a slight decrease in language modeling performance on math and coding \[5\].
> >
> > It is worth noting that our experimental results are consistent with the phenomena observed on well-recognized public benchmarks such as the Open LLM Leaderboard \[6\] and EvalPlus Leaderboard \[7\].
> >
> > **Reference:**
> > *\[1\] Ghosh S, Evuru C K R, Kumar S, et al. A Closer Look at the Limitations of Instruction Tuning\[J\]. arXiv preprint arXiv:2402.05119, 2024\.*
> > *\[2\] Mukherjee S, Mitra A, Jawahar G, et al. Orca: Progressive learning from complex explanation traces of gpt-4\[J\]. arXiv preprint arXiv:2306.02707, 2023\.*
> > *\[3\] Gunasekar S, Zhang Y, Aneja J, et al. Textbooks are all you need\[J\]. arXiv preprint arXiv:2306.11644, 2023\.*
> > *\[4\] Xu C, Sun Q, Zheng K, et al. Wizardlm: Empowering large language models to follow complex instructions\[J\]. arXiv preprint arXiv:2304.12244, 2023\.*
> > *\[5\] Ouyang L, Wu J, Jiang X, et al. Training language models to follow instructions with human feedback\[J\]. Advances in neural information processing systems, 2022, 35: 27730-27744.*
> > *\[6\] https://huggingface.co/spaces/open-llm-leaderboard-old/open\_llm\_leaderboard*
> > *\[7\] https://evalplus.github.io/leaderboard.html*

---

> > > ### Comment · Reviewer_h7cZ · 2024-11-26
> > >
> > > Thanks for your response, which resolves part of my concerns. But I'm still not convinced of the novelty (W1) and scalability (W2) of this method.
> > >
> > > For W1, compared with previous works, GLAN takes the same idea of knowledge structure for data synthesis. The methodology of CAMLE (biology/physics/chemistry) and MathScale can also extend to other disciplines, and there are also many works that do not require seed topics, such as MAmmoTH2 and MAGPIE.
> > >
> > > For W2, if human evaluation is only applied at the discipline level, it is unclear how to ensure the knowledge coverage of each discipline, the core contribution as the authors claim. Similarly, if the teacher model changes to Llama3, when the world knowledge (learned from pre-training) cannot cover the whole discipline, the constructed knowledge structure and generated QA samples can also be incomplete or even incorrect. There seems a trade-off between the quality and scalability of the proposed methodology.

---

> > > > ### Author Response · Authors · 2024-11-27
> > > > **Further Response (Part 1/2)**
> > > >
> > > > Thanks for your response and we are glad to hear that our clarification addresses some concerns. Below, we would like to further address your concerns by providing detailed responses to each of your comments:
> > > >
> > > > **A. Regarding Novelty**
> > > > We would like to first clarify our differences from the mentioned works:
> > > >
> > > > **Camel \[1\]**:
> > > > We acknowledge that Camel enhances problem-solving abilities through multi-agent role-playing. However, our research problem and method differs significantly from Camel,. While Camel generates data related to task resolution as a byproduct, there is no evidence that training on the generated data enhances the model's domain knowledge capabilities.
> > > >
> > > > **MathScale \[2\]:**
> > > >  "MathScale can also extend to other disciplines" is not always true. We believe MathScale can be extended to disciplines with sufficient seed questions/instructions.
> > > >
> > > > MathScale expands upon a set of high-quality seed instructions (20k math questions in their work) by generating new instructions based on a concept graph. This concept graph is constructed using co-occurrence statistics of concepts mined from the seed instructions. However, certain domains, such as nutrition, fishing, and textile, lack a sufficient number of high-quality seed instructions. Without an adequate quantity of seeds, it becomes challenging to ensure the quality of the concept graph, which in turn impacts the quality and diversity of the generated instructions.
> > > >
> > > > Our method does not need any seeds, and we have demonstrated that GLAN can successfully generate high quality and diverse  instructions across 126 different disciplines with good results (see Section 3 for details).
> > > >
> > > > **MAmmoTH2 \[3\]:**
> > > > Thanks for mentioning this great work. MAmmoTH2 discovers naturally existing instruction data from the web, but its coverage and diversity are influenced by the web corpus, making it prone to overlooking niche subjects. Additionally, the quality of the instruction pairs heavily depends on the quality of the web document classifier, which is non-trivial to build and requires significant human effort.
> > > >
> > > > Moreover, the quantity of instructions generated by MAmmoTH2 is limited by the number of filtered web pages. In contrast, our approach, through the combination of topics and key concepts, can generate 10 million or even more instructions.
> > > >
> > > >
> > > > **MAGPIE \[4\]:**
> > > > Although seed topic words are not required, MAGPIE requires model weights and the chat template words, which is not feasible for commercial APIs like GPT-4 and Claude that do not support inputting empty strings or chat control tokens.
> > > >
> > > > As illustrated in Figure 9 and Figure 10 of the MAGPIE paper, the task categories of the generated instruction-following data are predominantly information seeking (over 60%) and creative writing, with topics primarily centered around daily dialogue themes (such as story writing and planning). In contrast, GLAN focuses on general knowledge, encompassing a wide range of expert-level knowledge and textbook exercises, thereby providing instruction-tuning data with enhanced knowledge depth and breadth.
> > > >
> > > > Compared to related works, our method offers unique advantages in extending to general domains because:
> > > >
> > > > 1. GLAN is a general method for generating large-scale instruction data across diverse domains, applicable to a wide range of disciplines and tasks. GLAN uses a pre-curated taxonomy of human knowledge and capabilities to systematically and automatically generate large-scale instruction data across all disciplines. (Section 2.1 & 3.1)
> > > > 2. GLAN is scalable, capable of producing instructions on a massive scale using large language models (LLMs) with minimal human effort. (Section 2.4 & 3.1)
> > > > 3. GLAN is customizable, allowing for easy addition of new nodes to the taxonomy without the need to regenerate the entire dataset, enabling flexibility in adapting to new domains or requirements. (L104-107)
> > > > 4. GLAN incorporates a comprehensive syllabus into prompts (not based on single keywords), and thus can generate more challenging questions and high-quality answers. Extensive experiments on LLMs show GLAN's effectiveness across multiple dimensions, including mathematical reasoning, coding, academic exams, and logical reasoning, without requiring task-specific training data. (Table 1 & 2 & 4\)
> > > >
> > > > It is worth mentioning that **most of the mentioned works are concurrent with ours, with MAmmoTH2, MathScale and MAGPIE even being published later on arXiv**. We hope the above clarifications will allow you to reconsider the novelty assessment of our work.

---

> > > > > ### Author Response · Authors · 2024-11-27
> > > > > **Further Response (2/2)**
> > > > >
> > > > > **B. Regarding Scalability**
> > > > >
> > > > > Achieving the diversity of each discipline automatically is a key contribution of our method. There seems to be a misunderstanding—we do not need to manually ensure that every single knowledge point within each discipline is covered (which would be nearly impossible even for humans).  Our human verification is **at the discipline level** (126 disciplines in total, listed in supplementary materials), therefore, the cost of human verification is low and is **only required in the initial step** when constructing the taxonomy. **Scaling up does not require any additional human verification.**
> > > > >
> > > > > Our approach focuses on automatically controlling diversity, thereby ensuring wide coverage.
> > > > > To ensure the knowledge coverage of each discipline, for each discipline $d$, we extract a list of subjects using GPT-4 as a subject generator (Section 2.2). We then gather subject details (name, grade level, sub-topics) from GPT-4 completions and introduce the syllabus generator (Section 2.3) to break down the subject into smaller, hierarchical units. This approach ensures wide coverage (though not necessarily exhaustive) by outlining class sessions, sub-topics, and key concepts, providing a structured framework for high-quality questions.
> > > > >
> > > > > In fact, the query cost for ensuring knowledge coverage is minimal, as we only need to generate the syllabus. Using GPT-4 for this task incurs relatively few tokens, and we have already pre-generated and will release these syllabi. If an open-source model like Llama 3 is used, it can still use the syllabus and only generate questions and answers. According to Table 2 in Llama 3's technical report \[5\], Llama 3.1 70B performs comparably to GPT-4 Turbo 0125, which is stronger than the GPT-4-0613 model used in our experiments. In some tasks, Llama 3.1 70B even outperforms GPT-4 Turbo 0125\.
> > > > >
> > > > > **Reference:**
> > > > > *\[1\] Li G, Hammoud H, Itani H, et al. Camel: Communicative agents for" mind" exploration of large language model society\[J\]. Advances in Neural Information Processing Systems, 2023, 36: 51991-52008.*
> > > > > *\[2\] Tang Z, Zhang X, Wang B, et al. Mathscale: Scaling instruction tuning for mathematical reasoning\[J\]. arXiv preprint arXiv:2403.02884, 2024\.*
> > > > > *\[3\] Yue X, Zheng T, Zhang G, et al. Mammoth2: Scaling instructions from the web\[J\]. arXiv preprint arXiv:2405.03548, 2024\.*
> > > > > *\[4\] Xu Z, Jiang F, Niu L, et al. Magpie: Alignment Data Synthesis from Scratch by Prompting Aligned LLMs with Nothing\[J\]. arXiv preprint arXiv:2406.08464, 2024\.*
> > > > > *\[5\] Dubey A, Jauhri A, Pandey A, et al. The llama 3 herd of models\[J\]. arXiv preprint arXiv:2407.21783, 2024\.*

---

> > > > > > ### Comment · Reviewer_h7cZ · 2024-11-29
> > > > > >
> > > > > > Thanks for your further response, which clarifies the difference between your method and other related works. However, the core idea of GLAN is still exhaustively distilling GPT-4. It is not novel, and the coverage and quality of extracted knowledge structure are constrained by GPT-4's incomplete world knowledge (although it is the most powerful LLM), let alone weaker models like Llama3. I don't think it is a valuable idea to costly distill GPT-4 but will never exceed it. Overall, I tend to keep my score.

---

### Official Review · Reviewer_37wn · 2024-10-30

**Soundness:** 3
**Presentation:** 3
**Contribution:** 2
**Rating:** 5
**Confidence:** 3

**Summary:**

The paper presents GLAN (Generalized Instruction Tuning for Language Models), a scalable framework for generating synthetic instruction data to fine-tune Large Language Models (LLMs). GLAN's approach diverges from traditional methods by using a taxonomy of human knowledge and capabilities instead of seed examples or existing datasets. Inspired by educational structures, GLAN constructs synthetic instructions across disciplines by defining fields, sub-fields, and subjects, which are further detailed in-class sessions with key concepts. Using GPT-4 and GPT-3.5, the system generates extensive synthetic instruction datasets, demonstrating high performance across benchmarks.

**Strengths:**

1. This paper proposes a general method for generating synthetic instruction-tuning data, which can be easily scaled up to millions of samples.
2. The paper is well-written and easy to follow.
3. The performance of the model trained by this method is strong and promising.

**Weaknesses:**

1. The novelty of this paper might be limited.
[1] Enhancing Chat Language Models by Scaling High-quality Instructional Conversations.
This paper tries to first construct questions about worlds, and then generate instruction-tuning data based on them.
[2] Instruction Tuning with Human Curriculum.
This paper tries to construct datasets similarly from the education perspective.
[3] A Survey on Knowledge Distillation of Large Language Models.
It seems that in this paper, there is a whole subsection “KD Algorithms-Knowledge-Curation” that is generating data (almost) from scratch. Do these papers relate to your method?
I think at least these papers should be discussed.
2. Although this paper mentioned the scalability of this method, it seems that it largely depends on high-cost APIs, especially when millions of API calls are required.
3. The dataset proposed in this paper seems not to contain multi-turn conversations, which are crucial for many practical applications.

**Questions:**

1. Is the human evaluation really necessary? The involvement of humans might make the process hard to scale up automatically.

---

> ### Author Response · Authors · 2024-11-24
> **Response to Reviewer 37wn**
>
> Thank you for your review and the valuable feedback on our submission.
>
> **W1: Concern about novelty and related works**
>
> The novelty of our paper may appear limited compared to existing works such as [1], [2], and [3]. However, we aim to clarify that our method, GLAN, distinguishes itself by recursively decomposing high-level concepts into smaller, more granular elements. This recursive breakdown, unlike previous approaches, enhances the diversity and richness of synthesized instructions. By breaking down topics into their constituent subtopics, GLAN facilitates efficient knowledge distillation from Large Language Models (LLMs) to Small Language Models (SLMs). We consider prior works as special cases of GLAN, with a tree height of 1.
>
> Additionally, while [4] shares similarities with our approach in using syllabus to prompt LLMs, our method stands out by generating syllabus autonomously, broadening coverage and diversity of the synthesized instructions.
>
>
> **W2: Concern about API cost**
>
> Regarding the high-cost API concern you raised, we would like to clarify that although our current implementation involves high-cost APIs like GPT4, the method is adaptable to open-source LLMs like Llama 3 [5]. We intend to include experiments using data generated by Llama 3 in the final paper to demonstrate versatility.
>
>
> **W3: Lack of multi-turn conversation data**
>
> Your observation regarding the absence of multi-turn conversations in our dataset is valid. We recognize the importance of multi-turn dialogue for practical applications and plan to address this in future work by incorporating multi-turn conversation data.
>
>
> **Q1: Necessity of human evaluation**
>
> Regarding the necessity of human evaluation, our decision to employ human assessment was primarily to refine the taxonomy for efficiency. While human involvement can pose scalability challenges, algorithmic approaches like automated pruning based on content overlapping of the leaf node’s syllabus could streamline this process, ensuring scalability without compromising data quality.
>
>
> We appreciate your feedback and will incorporate these clarifications and expansions into the final paper.
>
>
> **Reference:**
>
> [1] Phi-2: The surprising power of small language models
>
> [2] Enhancing Chat Language Models by Scaling High-quality Instructional Conversations
>
> [3] Baize: An open-source chat model with parameter-efficient tuning on self-chat data*
>
> [4] Instruction Tuning with Human Curriculum
>
> [5] https://ai.meta.com/blog/meta-llama-3/

---

> > ### Comment · Reviewer_37wn · 2024-11-25
> >
> > Thanks for your reply.
> >
> > For W1, I don't think the idea of "breaking down topics into their constituent subtopics" has the necessary novelty and contribution for this year's ICLR conference.
> > For W2, the use of open-source models reduces the API costs but introduces more computation costs. And the performance of using open-source models is unknown.
> > For W3, it is indeed the limitation of the current method.
> >
> > Thus, I intend to keep my original score. Thank you.

---

> > > ### Author Response · Authors · 2024-11-28
> > > **Further Response (Part 1/2)**
> > >
> > > Thank you for your reply.
> > >
> > > **For W1 about the novelty, we'd like to add some complementary comments.**
> > >
> > > UltraChat[1] firstly query LLM to generate 30 topics that encompass various aspects of daily lives. Then, it generates 30-50 subtopics/concepts for each topic and asks LLM to generate questions based on each subtopics/concepts iteratively. Different from UltraChat, our GLAN generates questions based on the syllabus synthesized by LLM for each topic. The generated instructions are question-answering pairs of homework, which contain more educational value (e.g. clear explanations, step-by-step reasoning etc.).  As demonstrated in the previous study [2], the data with more education value benefits the downstream performance of LLM trained on such a dataset.
> > >
> > > [3] firstly collect real-world educational curricula from university catalogs and the Cambridge IGCSE curriculum, covering 45 subjects. Then they extract a set of concepts for each course description and synthesize instructions based on the extracted concepts with LLM. Different from [3], our GLAN generates syllabus (course description) autonomously with LLM. Since the syllabus is not limited to existing documents, the generated data would cover broader scope and more diverse topics.
> > >
> > > **Reference:**
> > >
> > > *[1] Ding, Ning, et al. "Enhancing Chat Language Models by Scaling High-quality Instructional Conversations." Proceedings of the 2023 Conference on Empirical Methods in Natural Language Processing. 2023.*
> > >
> > > *[2] Wettig, Alexander, et al. "QuRating: Selecting High-Quality Data for Training Language Models." Forty-first International Conference on Machine Learning.*
> > >
> > > *[3] Lee, Bruce W., Hyunsoo Cho, and Kang Min Yoo. "Instruction Tuning with Human Curriculum." Findings of the Association for Computational Linguistics: NAACL 2024. 2024.*
> > >
> > > **For w2 about the high API cost:**
> > >
> > > There are several papers about instruction synthesis, such as Orca2 [1] and MAmmoTH2 [2], which synthesize around 10 million questions with LLM. Our GLAN synthesize data with a similar scale.
> > >
> > > Furthermore, achieving the diversity of each discipline automatically is a key contribution of our method. To ensure the knowledge coverage, for each discipline, we extract a list of subjects using GPT-4 as a subject generator (Section 2.2). We then gather subject details (name, grade level, sub-topics) from GPT-4 completions and introduce the syllabus generator (Section 2.3) to break down the subject into smaller, hierarchical units. This approach ensures wide coverage (though not necessarily exhaustive) by outlining class sessions, sub-topics, and key concepts, providing a structured framework for high-quality questions.
> > >
> > > In fact, the query cost for ensuring knowledge coverage of GLAN is minimal, as we only need to generate the syllabus. Using GPT-4 for this task incurs relatively few tokens, and we have already pre-generated and will release these syllabi. If an open-source model like Llama 3 is used, it can still use the syllabus and only generate questions and answers. According to Table 2 in Llama 3's technical report [3], Llama 3.1 70B performs comparably to GPT-4 Turbo 0125, which is stronger than the GPT-4-0613 model used in our experiments. In some tasks, Llama 3.1 70B even outperforms GPT-4 Turbo 0125.
> > >
> > > **Reference:**
> > >
> > > *[1] Mitra, Arindam, et al. "Orca 2: Teaching small language models how to reason." arXiv preprint arXiv:2311.11045 (2023).*
> > >
> > > *[2] Yue, Xiang, et al. "MAmmoTH2: Scaling Instructions from the Web." Advances in Neural Information Processing Systems, 2024*
> > >
> > > *[3] Dubey A, Jauhri A, Pandey A, et al. The llama 3 herd of models[J]. arXiv preprint arXiv:2407.21783, 2024.*

---

> > > > ### Author Response · Authors · 2024-11-28
> > > > **Further Response (Part 2/2)**
> > > >
> > > > **For w3 about lack of multi-turn conversational data:**
> > > >
> > > > Although our dataset does not contain multi-turn conversational data, we would like to clarify that the research of constructing multi-turn conversational data is orthogonal to our GLAN. Specifically, we can utilize the GLAN pipeline to create initial instruction-response pairs for the first turn. To expand these into multi-turn interactions, we can integrate methods from the field of generating multi-turn conversational data for subsequent turns, such as the one proposed in [4], by adding another role of LLM aiming at proposing questions based on the conversational history. There are many existing works on data synthesis focusing on single-turn data, such as Wizardlm[1], Self-Instruct [2] and Unnatural Instructions [3] etc. These works would also be extended to multi-turn conversational data with this “two-agents role playing” technique.
> > > >
> > > > **Reference**
> > > >
> > > > *[1] Xu, Can, et al. "Wizardlm: Empowering large language models to follow complex instructions." arXiv preprint arXiv:2304.12244 (2023).*
> > > >
> > > > *[2] Wang, Yizhong, et al. "Self-Instruct: Aligning Language Models with Self-Generated Instructions." Proceedings of the 61st Annual Meeting of the Association for Computational Linguistics (Volume 1: Long Papers). 2023.*
> > > >
> > > > *[3] Honovich, Or, et al. "Unnatural Instructions: Tuning Language Models with (Almost) No Human Labor." Proceedings of the 61st Annual Meeting of the Association for Computational Linguistics (Volume 1: Long Papers). 2023.*
> > > >
> > > > *[4] Ding, Ning, et al. "Enhancing Chat Language Models by Scaling High-quality Instructional Conversations." Proceedings of the 2023 Conference on Empirical Methods in Natural Language Processing. 2023.*
> > > >
> > > > Overall, Compared to related works, our method offers unique advantages in extending to general domains because:
> > > > 1. GLAN is a general method for generating large-scale instruction data across diverse domains, applicable to a wide range of disciplines and tasks. GLAN uses a pre-curated taxonomy of human knowledge and capabilities to systematically and automatically generate large-scale instruction data across all disciplines. (Section 2.1 & 3.1)
> > > > 2. GLAN is scalable, capable of producing instructions on a massive scale using large language models (LLMs) with minimal human effort. (Section 2.4 & 3.1)
> > > > 3. GLAN is customizable, allowing for easy addition of new nodes to the taxonomy without the need to regenerate the entire dataset, enabling flexibility in adapting to new domains or requirements. (L104-107)
> > > > 4. GLAN incorporates a comprehensive syllabus into prompts (not based on single keywords), and thus can generate more challenging questions and high-quality answers. Extensive experiments on LLMs show GLAN's effectiveness across multiple dimensions, including mathematical reasoning, coding, academic exams, and logical reasoning, without requiring task-specific training data. (Table 1 & 2 & 4)
> > > >
> > > > We appreciate your feedback and will incorporate these clarifications and expansions into the final paper.

---

### Official Review · Reviewer_7dnk · 2024-11-02

**Soundness:** 4
**Presentation:** 4
**Contribution:** 3
**Rating:** 5
**Confidence:** 4

**Summary:**

The submitted paper introduces GLAN as a method for instruction tuning of LLMs with synthetic data generated without seed examples or pre-existing datasets. The proposed approach structures synthetic data based on a taxonomy that decomposes human knowledge systematically, inspired by the educational system.

Key Contributions:
1. Novel Data Generation Approach: This intuitive and inspiring taxonomy-based instruction generation method ensures broad coverage across various knowledge domains.
2. Scalability and Customization: GLAN's structure enables the easy addition of new fields by incorporating new nodes into the taxonomy, making it a scalable and adaptable solution for future advancements in LLMs.

**Strengths:**

Originality: The paper introduces a novel, taxonomy-driven approach to instruction tuning inspired by structured education systems, moving beyond the traditional reliance on seed examples and transforming data creation into a structured method rather than uncontrolled expansion. By systematically decomposing human knowledge, GLAN enables broader and more versatile LLM tuning in an organized manner, marking a creative adaptation of structured learning for AI.

Quality: The methodology is detailed and robust, employing models like GPT-4 and GPT-3.5 to ensure quality and incorporating human oversight to reduce hallucination. Extensive experiments demonstrate strong results across multiple benchmarks, highlighting the model’s adaptability and thoughtful experimental design.

Clarity: The paper is well-organized, with each component of the GLAN process clearly explained. A detailed appendix provides additional information for deeper exploration. Figures and tables effectively illustrate benchmark results and comparisons, making the model’s strengths easy to understand.

Significance: By supporting easy expansion into diverse fields and facilitating curriculum-based learning, GLAN addresses the demand for larger, more capable models. It enables the creation of expansive and diverse datasets, establishing a new approach to improving dataset quality. GLAN’s scalable, customizable approach to synthetic data generation holds significant potential for advancing multi-domain LLMs and enhancing general model capabilities.

**Weaknesses:**

Human Effort and Taxonomy Standardization: The taxonomy creation phase includes a human verification step, but the exact extent of human involvement and the criteria for removing items from the taxonomy are not clearly defined. It would be helpful to clarify the standard used for verifying and potentially removing fields, sub-fields, or disciplines to improve the reproducibility of the approach. Establishing more detailed criteria for this step could ensure consistency and transparency, as well as allow other researchers to adapt this approach efficiently.

High Computational Cost and Open-Source Model Viability: The paper relies heavily on closed-source models like GPT-4 and GPT-3.5, which incur significant costs. It remains unclear whether switching to open-source models would provide comparable performance. Investigating the performance trade-offs when using state-of-the-art open-source models, such as LLaMA 3, could offer a more cost-effective alternative while maintaining accuracy. Including a discussion on the feasibility of using open-source models, with supporting experiments if possible, would improve accessibility for researchers with limited resources.

**Questions:**

1. Clarification on Human Effort in Taxonomy Creation:
- Question: Could you elaborate on the level of human involvement in this pipeline? Specifically, how much human intervention is required, and what criteria or standards are used to decide which fields, sub-fields, or disciplines to retain or remove? Additionally, could you explain the underlying rationale for determining where human involvement is necessary versus where it is not?
- Suggestion: Providing more details on the standards for pruning the taxonomy would enhance transparency and reproducibility. Examples of cases where human verification led to the removal or modification of specific fields would also help clarify the reasoning behind the taxonomy’s final structure, offering insights into the decision-making process in this stage.

2. Exploring Open-Source Model for Data Generation:
- Question: If data generation in GLAN were performed using open-source models, such as LLaMA 3, how would its performance levels compare to those obtained with GPT-4 and GPT-3.5?
- Suggestion: Given the high computational cost associated with using GPT-4 and GPT-3.5, conducting an experiment with open-source models would add valuable insights. A comparative analysis of GLAN’s performance across various model sizes and types could illustrate its adaptability and provide practical guidance for researchers with different resource constraints. This would be especially useful in determining GLAN’s cost-effectiveness and accessibility.

---

> ### Author Response · Authors · 2024-11-24
>
> Thank you for your questions and insightful suggestions regarding our work!
>
> **Q1. Clarification on Human Effort in Taxonomy Creation**
>
> Human intervention is only required at the **initial stage** of building the taxonomy.
>
> Specifically, we begin by prompting GPT-4 multiple times with the query "list all fields of human knowledge and capabilities", resulting in the collection of hundreds of disciplines. To manage the presence of duplicated or similar disciplines, a **majority voting** process is conducted among 3 human annotators to determine which disciplines should be retained. Once this initial curation is completed, the subsequent generation steps within GLAN are fully automated to minimize the need for human involvement.
>
> **Q2. Exploring Open-Source Models for Data Generation**
>
> You are correct—GLAN is designed to be model-agnostic, meaning it can operate with both closed-source models (e.g., GPT-4/GPT-3.5) and strong open-source models (e.g., LLaMA-3 72B). In our experiments, we chose to use closed-source models to best demonstrate GLAN’s potential at that time. However, we anticipate that GLAN can achieve comparable results when applied to high-performing open-source models, making it more applicable.

---

> > ### Comment · Reviewer_7dnk · 2024-11-29
> > **Thank you for your response.**
> >
> > Thank you for your response. My concerns are well addressed. I'd like to keep the rating.

---

### Meta-Review · Area_Chair_1i38 · 2024-12-21

**Metareview:**

The paper introduces GLAN, a method for tuning LLMs using synthetic data generated from a taxonomy of human knowledge, aimed at creating diverse instructional data and enhancing LLM performance.

While the structured approach to generating synthetic data is compelling, concerns about the method's novelty remain due to similarities with existing approaches. Additionally, reliance on proprietary models like GPT-4 and the need for human verification pose scalability and reproducibility challenges, limiting its broader applicability.

The experimental evaluation lacks extensive baseline comparisons and reveals shortcomings in the generated data. Given these issues, the paper does not meet the criteria for acceptance.

**Additional Comments On Reviewer Discussion:**

During the rebuttal period, reviewers raised concerns about the novelty and scalability of GLAN. Reviewer 37wn and Reviewer h7cZ questioned the originality of constructing knowledge structures for data synthesis, noting similarities to prior work. They also highlighted scalability issues, particularly the reliance on high-cost proprietary models like GPT-4 for data generation, and expressed doubts about the method's performance with open-source models. Reviewer 1yM3 and Reviewer 7dnk* aligned with the other reviewers, concluding that the paper lacks sufficient novelty to meet the acceptance threshold. All reviewers agreed that while the approach has potential, it does not represent a significant advancement over existing methods.

In response, the authors emphasized GLAN's recursive decomposition of high-level concepts without seed examples, arguing that it enhances instruction diversity. They also claimed the method is scalable and model-agnostic, producing quality results with open-source models, though they acknowledged performance might vary.

Despite these clarifications, the reviewers maintained their stance, agreeing that the paper does not meet the criteria for acceptance due to limited novelty and unresolved scalability concerns.

---

### Decision · Program_Chairs · 2025-01-22

Reject